# GPTAQ: Efficient Finetuning-Free Quantization for Asymmetric Calibration

Yuhang Li [1]   Ruokai Yin [1]   Donghyun Lee [1]   Shiting Xiao [1]   Priyadarshini Panda [1]

## Abstract

We introduce GPTAQ, a novel finetuning-free quantization method for compressing large-scale transformer architectures. Unlike the previous GPTQ method, which independently calibrates each layer, we always match the quantized layer's output to the exact output in the full-precision model, resulting in a scheme that we call *asymmetric calibration*. Such a scheme can effectively reduce the quantization error accumulated in previous layers. We analyze this problem using optimal brain compression to derive a close-formed solution. The new solution explicitly minimizes the quantization error as well as the accumulated asymmetry error. Furthermore, we utilize various techniques to parallelize the solution calculation, including channel parallelization, neuron decomposition, and Cholesky reformulation for matrix fusion. As a result, GPTAQ is easy to implement, simply using 20 more lines of code than GPTQ but improving its performance under low-bit quantization. Remarkably, *on a single GPU*, we quantize a 405B language transformer as well as EVA-02—the rank first vision transformer that achieves 90% pretraining Imagenet accuracy. Code is available at Github.

## 1. Introduction

The emergence of transformer architectures (Vaswani, 2017) has led to unprecedented scaling in model sizes and computational demands in both vision and language domains. In computer vision, models like ViT-G/14 (Zhai et al., 2022) encompass 2 billion parameters and require 2860 GFLOPs—700 times that of ResNet-50—for processing a single image. Language models have scaled even further, with architectures like LLaMA-3-405B (Meta, 2024) reaching hundreds of billions of parameters. This scale poses

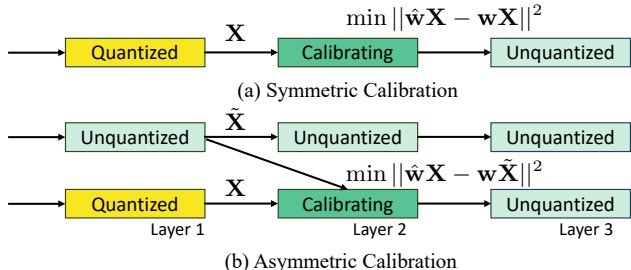

Figure 1. Calibration pipeline in the symmetric way (GPTQ) and the asymmetric way (GPTAQ).

significant deployment challenges across all computing platforms, from servers to edge devices. Moreover, transformers enable multi-modal architectures, which combine multiple transformer architectures and make it challenging to operate under resource-constrained environments.

Quantization (Gholami et al., 2022) has emerged as a promising approach to address these challenges by reducing the precision of weights and activations, thereby accelerating both memory access and computation. However, current state-of-the-art quantization methods often rely on model fine-tuning. These approaches (Shao et al., 2023; Liu et al., 2024) update parameters through gradient descent, a process that becomes increasingly challenging with model size. For instance, Malinovskii et al. (2024) requires 8 A100 GPUs running for 8 days to fine-tune a 70B parameter language model. This computational burden becomes particularly problematic given the numerous possible quantization configurations, including bit formats, symmetry choices, etc., making fine-tuning-based quantization less efficient.

This work focuses on *finetuning-free quantization*—an approach that circumvents gradient-based optimization in favor of forward-pass calibration computation. Several methods have emerged in recent literature (Lin et al., 2023; Dettmers et al., 2022; Nagel et al., 2019; Liu et al., 2021). Among these approaches, GPTQ (Frantar et al., 2022)[1] has distinguished itself through a particularly effective combination of speed and accuracy, leading to its widespread adoption across the machine learning community, with over 5,321 quantized transformers now available on the Hugging-

---

[1]Department of Electrical Engineering, Yale University. Correspondence to: Yuhang Li <yuhang.li@yale.edu>.

*Proceedings of the 42nd International Conference on Machine Learning*, Vancouver, Canada. PMLR 267, 2025. Copyright 2025 by the author(s).

[1]While the authors used the name "OPTQ" in the final camera-ready paper, we use "GPTQ" throughout this work as it has become the more widely recognized name in the broader community.

face Hub (Huggingface, 2025). The method's broad API support and straightforward implementation have made it the de facto standard for quantizing emerging transformer architectures, enabling rapid deployment.

In this work, we identify a crucial problem in GPTQ, which we call symmetric calibration. The symmetric calibration problem arises from the fact that each layer optimizes the objective of $||\hat{\mathbf{w}}\mathbf{X} - \mathbf{w}\mathbf{X}||_F^2$ where the input $\mathbf{X}$ is coming from the previous quantized layer's output. This input poses a deviation from the original model's input activation $\tilde{\mathbf{X}}$, as shown in Fig. 1. Our objective involves the input activation from the full precision model, which we call asymmetric calibration: In this work, we tackle asymmetric calibration by proposing GPTAQ, where A stands for input activation asymmetry. Our contribution includes:

1. We analyze the asymmetric calibration and show that the optimal update to weights needs to account for quantization error, inverse Hessian, as well as input deviation.
2. We propose to execute the asymmetric calibration via several modifications, including efficiently parallelizing all output channels and decomposing the residual error into each channel dimension to leverage efficiency.
3. Using the above method, our proposed GPTAQ only requires 20 lines of more code than GPTQ and improve its performance on both vision and language transformers. GPTAQ effectively quantizes extremely large transformers, including LLaMA3.1-405B and EVA-02.

## 2. Related Work

**Finetuning-Free Quantization.** Finetuning-free quantization gathers particular interest as it can immediately export quantized checkpoints given the massiveness of pre-trained large models. A classic finetuning-free approach is to correct the distribution using bias and scales, (Banner et al., 2019; Lin et al., 2023; Nagel et al., 2019). Moreover, Lin et al. (2023); Xiao et al. (2023); Wei et al. (2022b; 2023) have proposed to reduce the outlier problem in language transformers. Other approaches involve architecture modifications like channel splitting and merging (Zhao et al., 2019; Liu et al., 2023a). Recently, incoherence processing (Chee et al., 2023) has been proposed to transform the weight distribution into a more quantization-friendly one. Tseng et al. (2024) adopts Hadamard transforms to improve transform efficiency and Ashkboos et al. (2024) further extends it with an online operation to transform activations. Heterogeneous treatment, e.g., Dettmers et al. (2022); Huang et al. (2024a;b), allocates more quantization spaces (higher precision) for salient entries, which, however, requires complex hardware design.

The precursor of our method, GPTQ (Frantar et al., 2022) optimizes the weight elements using closed-form solutions and thus does not require backpropagation. Both our method and GPTQ can be combined with above mentioned methods.

**Finetuning-based Quantization.** Fintuning-based quantization costs more computation resources to compute forward and backward propagation. Full network finetuning (Malinovskii et al., 2024; Liu et al., 2024; Dettmers et al., 2023; Wang et al., 2023) can achieve good performance but they are restricted to smaller models. Local finetuning approaches can alleviate this problem (Nagel et al., 2020; Hubara et al., 2020; Li et al., 2021; Wei et al., 2022a; Shao et al., 2023; Tseng et al., 2024), nevertheless, the time and resources needed are still a lot more than finetuning-free approaches.

**Optimal Brain Surgeon (OBS).** It was originally applied to small networks with hundreds of weights (LeCun et al., 1989; Hassibi et al., 1993). Efforts have been made to reduce the estimation complexity of Hessian on larger models, like Fisher approximation (Singh & Alistarh, 2020), K-FAC approximation (Dong et al., 2017). Our approach extends the Optimal Brain Compression (Frantar & Alistarh, 2022) which uses Gram layer input as the Hessian matrix to perform layer-wise quantization. The key difference is the introduction of a correction term in the input for ground truth target, leading to an asymmetric calibration framework.

## 3. Background

### 3.1. Notations

We adopt row-vector notation throughout this paper. Vectors and matrices are denoted by bold lowercase and uppercase letters respectively. For instance, a linear operation between a weight vector and input activations is expressed as: $\mathbf{y} = \mathbf{w}\mathbf{X}$ where $\mathbf{w} \in \mathbb{R}^{1 \times n}$ represents a row of weights (output channel) and $\mathbf{X} \in \mathbb{R}^{n \times k}$ denotes the input activation matrix, with $n$ being the number of input neurons. Quantization is represented as $\hat{\mathbf{w}} = \text{quant}(\mathbf{w})$. For indexing, we use subscripts to denote specific elements or subsets of vectors and matrices. Notably, negative indices indicate the removal of a neuron (corresponding to a row in $\mathbf{X}$). For example, $\mathbf{X}_{-1} \in \mathbb{R}^{(n-1) \times k}$ represents the input activation matrix with its first input neuron removed.

### 3.2. OBQ & GPTQ

Converting floating-point model parameters $\mathbf{W} \in \mathbb{R}^{m \times n}$ from FP16 to integer representations $\hat{\mathbf{W}}$ demands a calibration process to preserve model behavior. This calibration is formalized in the Optimal Brain Quantization (OBQ) framework, which GPTQ implements efficiently for large models. At its core, OBQ calibration process minimizes the difference between the original and quantized layer outputs:

$$\min_{\Delta\mathbf{w}} ||(\mathbf{w} + \Delta\mathbf{w})\mathbf{X} - \mathbf{w}\mathbf{X}||_F^2, \text{ s.t. } \Delta\mathbf{w} = \hat{\mathbf{w}} - \mathbf{w}. \quad (1)$$

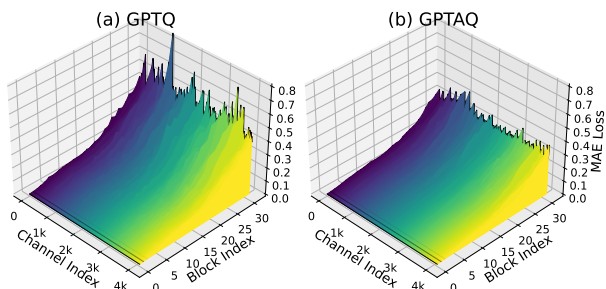

*Figure 2.* Visualization of input activation MAE loss ($|\widetilde{\mathbf{X}} - \mathbf{X}|$) when calibrating LLaMA3-8B using GPTQ and GPTAQ.

Using OBS framework (Hassibi et al., 1993), OBQ solves this optimization problem iteratively. For each weight $q$, it computes the optimal quantized value and corresponding adjustment:

$$q = \arg\min_q \frac{(\hat{\mathbf{w}}_q - \mathbf{w}_q)^2}{\mathbf{H}_{qq}^{-1}}, \quad \Delta\mathbf{w} = \frac{(\hat{\mathbf{w}}_q - \mathbf{w}_q)}{\mathbf{H}_{qq}^{-1}} \cdot (\mathbf{H}_{q,:}^{-1}). \tag{2}$$

where $\mathbf{H}^{-1} = (\mathbf{X}\mathbf{X}^\top)^{-1}$ represents the inverse Hessian matrix. After quantizing each weight, the inverse Hessian matrix is updated efficiently using Gaussian Elimination to exclude the quantized entry:

$$\mathbf{H}_{-q}^{-1} = (\mathbf{X}_{-q}\mathbf{X}_{-q}^\top)^{-1} = \left(\mathbf{H}^{-1} - \frac{\mathbf{H}_{:,q}^{-1}\mathbf{H}_{q,:}^{-1}}{\mathbf{H}_{qq}^{-1}}\right) \tag{3}$$

While OBQ provides a closed-form solution, its computational cost becomes prohibitive for billion-parameter transformer models. GPTQ (Frantar et al., 2022) addresses this limitation by enhancing parallelization and improving numerical stability through Cholesky Reformuation, making the approach practical and robust for models with hundreds of billions of parameters.

## 4. GPTAQ Methodology

We contend that a key limitation of GPTQ and the original OBQ lies in their symmetric treatment of input activations, that fails to account for how quantized layers progressively transform the activation patterns. This oversight, which becomes particularly acute as quantization proceeds through deeper layers, motivates our introduction of an *asymmetric calibration* framework designed to preserve the characteristics of the floating-point model's layer inputs $\widetilde{\mathbf{X}}$. The optimization objective becomes:

$$\min_{\Delta\mathbf{w}} ||(\mathbf{w} + \Delta\mathbf{w})\mathbf{X} - \mathbf{w}\widetilde{\mathbf{X}}||_F^2, \quad \text{s.t. } \Delta\mathbf{w} = \hat{\mathbf{w}} - \mathbf{w}, \tag{4}$$

In practice the difference $(\widetilde{\mathbf{X}} - \mathbf{X})$ can come from both activation quantization and weight quantization in previous layers. This deviation accumulates systematically through the network's depth—a phenomenon we show in Fig. 2(a),

where the activation MAE loss across transformer blocks continues to increase during GPTQ quantization, and thus reveals a fundamental bias in symmetric calibration. To address this challenge comprehensively, our work advances two primary contributions: first, a rigorous theoretical framework that optimally handles asymmetric calibration, and second, an efficient GPU implementation strategy that crystallizes these insights into the GPTAQ framework. We also show the MAE loss surfaces using GPTAQ in Fig. 2(b).

### 4.1. Optimal Framework

We now develop a solution to the asymmetric calibration problem defined in Eq. (4). Let us first consider optimization for a single output channel (one row of weights). We introduce $\mathbf{r}$ to represent the output activation residuals $\mathbf{w}\widetilde{\mathbf{X}} - \mathbf{w}\mathbf{X}$, allowing us to reformulate the objective as:

$$\min_{\Delta\mathbf{w}} ||\Delta\mathbf{w}\mathbf{X} - \mathbf{r}||_F^2, \quad \text{s.t. } \Delta\mathbf{w} = \hat{\mathbf{w}} - \mathbf{w}. \tag{5}$$

Our optimization strategy proceeds iteratively by:

1. Selecting a weight $\mathbf{w}_q$ for quantization,
2. Computing the optimal weight updates $\Delta\mathbf{w}$ to minimize the loss function.

For the $q$-th weight quantization, we introduce the constraint $\Delta\mathbf{w}\mathbf{e}_q^\top + \mathbf{w}_q - \hat{\mathbf{w}}_q = 0$, where $\mathbf{e}_q$ represents a unit vector with all elements set to 0 except for a 1 in the $q$-th position. This leads to a nested optimization problem:

$$\min_q \left\{ \min_{\Delta\mathbf{w}}\{||\Delta\mathbf{w}\mathbf{X} - \mathbf{r}||_F^2\} \text{ s.t. } \Delta\mathbf{w}\mathbf{e}_q^\top + \mathbf{w}_q - \hat{\mathbf{w}}_q = 0 \right\}. \tag{6}$$

To solve this constrained optimization problem, we formulate its Lagrangian:

$$L = ||\Delta\mathbf{w}\mathbf{X} - \mathbf{r}||_F^2 + \lambda(\Delta\mathbf{w}\mathbf{e}_q^\top + \mathbf{w}_q - \hat{\mathbf{w}}_q). \tag{7}$$

The optimal solutions are obtained by taking derivatives of the Lagrangian:

$$\begin{cases} \dfrac{\partial L}{\partial \Delta\mathbf{w}} = 2\Delta\mathbf{w}\mathbf{H} - 2\mathbf{r}\mathbf{X}^\top + \lambda\mathbf{e}_q \\ \dfrac{\partial L}{\partial \lambda} = \Delta\mathbf{w}\mathbf{e}_q^\top + \mathbf{w}_q - \hat{\mathbf{w}}_q \end{cases}, \tag{8}$$

and setting these derivatives to zero, which yields the optimal weight update $\Delta\mathbf{w}$:

$$\Delta\mathbf{w} = \frac{(\hat{\mathbf{w}}_q - \mathbf{w}_q)}{\mathbf{H}_{qq}^{-1}} \cdot (\mathbf{H}_{q,:}^{-1}) + \mathbf{r}\mathbf{X}^\top\mathbf{H}_{-q}^{-1}, \tag{9}$$

with the corresponding loss function:

$$L_q = \frac{(\hat{\mathbf{w}}_q - \mathbf{w}_q)^2}{\mathbf{H}_{qq}^{-1}} + \mathbf{r}\mathbf{r}^\top - \mathbf{r}\mathbf{X}^\top\mathbf{H}_{-q}^{-1}\mathbf{X}\mathbf{r}^\top$$
$$- 2\frac{(\hat{\mathbf{w}}_q - \mathbf{w}_q)}{\mathbf{H}_{qq}^{-1}}\mathbf{r}\mathbf{X}^\top\mathbf{H}_{:,q}^{-1}, \tag{10}$$

where $\mathbf{H}_{-q}^{-1}$, according to Eq. (3), represents the inverse Hessian matrix after applying Gaussian elimination to zero out the $q$-th row and column. The complete derivation is provided in Appendix A.1. The optimal framework proceeds iteratively by first computing $q = \arg\min_q L_q$, then quantizing the $q$-th weight to $\hat{\mathbf{w}}_q$, and finally updating the remaining full-precision weights by $\Delta\mathbf{w}$. This process continues until all weight elements are quantized.

However, this direct implementation faces significant efficiency challenges when applied to large transformer models, that stem not only from the computational cost of evaluating $L_q$ for each weight, but also from the way different optimal $q$ orderings across output channels prevent effective GPU parallelization. The computational burden is particularly severe as the process involves multiple $n \times n$ matrix multiplications for every single weight, becoming prohibitively expensive for large foundation models. Furthermore, each weight update necessitates re-estimation of the residual output $\mathbf{r}$. To address these limitations, we propose four optimization steps that achieve efficiency comparable to GPTQ, which we detail in the following section.

### 4.2. Efficient Solution

**Step 1, Arbitrary orders of handling $q$.** The massive parameter space of large-scale foundation models suggests an important optimization: we can process weights in an arbitrary order rather than strictly following the optimal sequence of $L_q$. When quantizing individual weight elements, the extensive weight space provides sufficient flexibility to compensate for quantization errors. This insight is supported by empirical evidence from GPTQ (Frantar et al., 2022), which demonstrates that following the optimal descending order of $L_q$ offers only marginal improvements over arbitrary ordering.

Processing weights in an arbitrary order yields an additional practical advantage: it enables parallel processing of all rows, efficiently utilizing GPU computational capabilities. We therefore adopt GPTQ's order of processing columns, which is sequentially from the first column to the last. For each $q = 1, 2, 3, \ldots, n$, we compute the weight updates $\Delta\mathbf{W}$ across all rows simultaneously using:

$$\Delta\mathbf{W} = \frac{(\hat{\mathbf{W}}_{:,q} - \mathbf{W}_{:,q})}{\mathbf{H}_{qq}^{-1}} \cdot (\mathbf{H}_{q,:}^{-1}) + \mathbf{R}\mathbf{X}^\top\mathbf{H}_{-q}^{-1}, \quad (11)$$

where $\mathbf{R} = \mathbf{W}\tilde{\mathbf{X}} - \mathbf{W}\mathbf{X}$ represents the output residuals across all rows (output channels). This formulation allows us to maintain a single copy of $\mathbf{H}_{-q}^{-1}$ as it remains constant across all rows for a given $q$.

While this approach enables parallel processing of rows, two significant computational challenges remain: the need to estimate new residuals $\mathbf{R}$ at each iteration, and the computational burden of the matrix multiplication $\mathbf{R}\mathbf{X}^\top\mathbf{H}_{-q}^{-1}$.

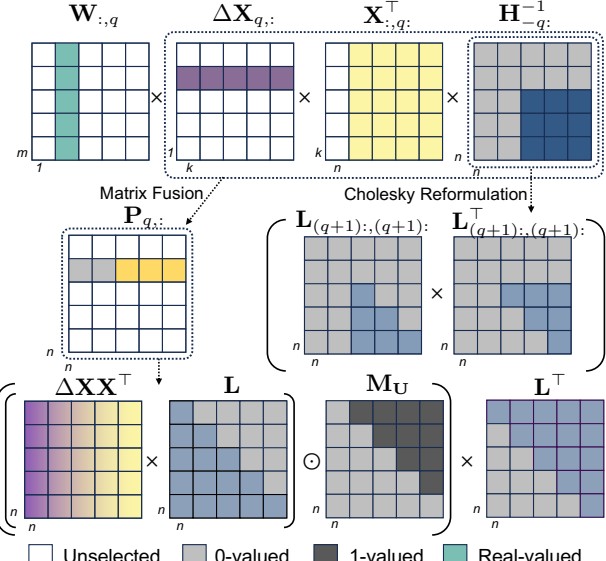

*Figure 3.* Computing paradigm of the second term for residual output error in $q = 2$ iteration. The inverse Hessian matrix is factorized by Cholesky Decomposition, furthermore, $\Delta\mathbf{X}_{q,:}\mathbf{X}^\top\mathbf{H}_{-q}^{-1}$ is fused to the $q$-th row of matrix $\mathbf{P}$, which can be computed in parallel. Dimensions are in the bottom left corner of each matrix.

These challenges should be addressed to achieve practical efficiency for large-scale models.

**Step 2, Efficient Residual Decomposition.** Another crucial efficiency consideration in our algorithm is the repeated computation of residuals $\mathbf{R}$. Although the term $\mathbf{R}\mathbf{X}^\top\mathbf{H}_{-q}^{-1}$ effectively reduces the gap between quantized and full-precision model outputs, the residual output $\mathbf{R}$ needs to be re-estimated at each new iteration. Denoting the residual error in input activation $\Delta\mathbf{X} = \tilde{\mathbf{X}} - \mathbf{X}$, the update formula is given by

$$\mathbf{R}_{\text{new}} = \mathbf{R}_{-q} - \Delta\mathbf{W}\Delta\mathbf{X}_{-q}, \quad (12)$$

which incurs a substantial computational cost, with complexity $O(mnk)$, where $k$ is the product of token length and number of calibration samples. The dependence on $k$ poses a particular challenge for large foundation models, where $k$ typically exceeds $m$ and $n$ by a factor of $10 \sim 50$ (for example on LLaMA2-7B the $k$ is $128 \times 2048$, which is $64\times$ greater than $n(4096)$), leading to prohibitive processing times.

We propose an alternative approach that eliminates the need to recalculate residual outputs $\mathbf{R}$. Our key insight is that residuals can be decomposed into $n$ individual components from different neurons:

$$\mathbf{R} = \mathbf{W}\Delta\mathbf{X} = \sum_{q=1}^{n} \mathbf{W}_{:,q}\Delta\mathbf{X}_{q,:}, \quad (13)$$

This decomposition enables a single estimation of $\mathbf{R}$ before

layer calibration. Hence, at $q$-th iteration, with the first $q-1$ columns quantized, we focus on quantizing the $q$-th column while minimizing its associated residual error component:

$$\min_{\Delta\mathbf{W}_{:,q:}} ||\Delta\mathbf{W}_{:,q:}\mathbf{X}_{q:,:} - \mathbf{W}_{:,q}\Delta\mathbf{X}_{q,:}||_F^2 \quad (14)$$
$$\text{s.t. } \Delta\mathbf{W}_{:,q}\mathbf{e}_q^\top + \mathbf{W}_{:,q} - \hat{\mathbf{W}}_{:,q} = \mathbf{0},$$

Similarly, the optimal weight update for the $q$-th iteration becomes:

$$\Delta\mathbf{W}_{:,q:} = \frac{(\hat{\mathbf{W}}_{:,q} - \mathbf{W}_{:,q})}{\tilde{\mathbf{H}}_{qq}^{-1}}\cdot(\tilde{\mathbf{H}}_{q,:}^{-1}) + \mathbf{W}_{:,q}\Delta\mathbf{X}_{q,:}\mathbf{X}_{:,q:}^\top\tilde{\mathbf{H}}_{-q}^{-1}. \quad (15)$$

where $\tilde{\mathbf{H}}^{-1} = \mathbf{H}_{-(q-1):}^{-1}$ is the inverse Hessian matrix eliminated by $(q-1)$ iterations, therefore $\tilde{\mathbf{H}}_{-q}^{-1} = \mathbf{H}_{-q:}^{-1}$. Notably, the first term in Eq. (15) is the same with GPTQ, while the second term is accounted for minimizing the residual error in $q$-th neuron and does not need re-estimation of residual error since $\mathbf{W}_{:,q}$ is already updated by previous $\Delta\mathbf{W}$. Fig. 3 gives an example of the second term when $q = 2$.

Furthermore, the residual decomposition yields significant computational advantages because we reduce the complexity of the second term by a factor of $n$, due to the fact that $\Delta\mathbf{X}_{q,:}\mathbf{X}_{:,q:}^\top\mathbf{H}_{-q:}^{-1} \in \mathbb{R}^{1\times(n-q)}$ is a row vector and $\mathbf{W}_{:,q}$ is a column vector, requiring only $\mathcal{O}(mn)$ complexity to compute the second term. We further optimize our algorithm by noting that $\Delta\mathbf{X}_{q,:}\mathbf{X}_{:,q:}^\top\mathbf{H}_{-q:}^{-1}$ is independent to the update of weights, thus we can compute each row in advance and store it in a matrix $\mathbf{P}$, such that the second term can be efficiently computed by $\mathbf{W}_{:,q}\mathbf{P}_{q,:}$ in $q$-th iteration.

**Step 3, Cholesky Reformulation.** The final challenge lies in computing $\mathbf{P}$ efficiently while maintaining numerical stability. These requirements arise from the need to perform Gaussian Elimination on the inverse Hessian $n$ times and compute matrix multiplications. As model size increases, repeated Gaussian Elimination operations accumulate numerical errors. We address this numerical instability through Cholesky decomposition of the inverse Hessian: $\mathbf{H}^{-1} = \mathbf{L}\mathbf{L}^\top$, where $\mathbf{L}$ represents the lower-triangular Cholesky factor. This enables efficient and stable computation of $\mathbf{H}_{-q:}^{-1}$ through the following lemma:

**Lemma 4.1.** *Given the Cholesky factor $\mathbf{L}$ for the full inverse Hessian matrix $\mathbf{H}^{-1}$, the inverse Hessian $\mathbf{H}_{-q:}^{-1} = (\mathbf{X}_{-q:}\mathbf{X}_{-q:}^\top)^{-1}$ is equivalent to $\mathbf{L}_{q+1:,q+1:}\mathbf{L}_{q+1:,q+1:}^\top$.*

The proof is provided in Appendix A.2. This formulation offers better numerical stability compared to Gaussian Elimination, requiring only matrix slicing and multiplication to obtain $\mathbf{H}_{-q:}^{-1}$. Consequently, we can reformulate $\mathbf{P}_{q,:}$ as:

$$\mathbf{P}_{q,:} = \Delta\mathbf{X}_{q,:}\mathbf{X}^\top\mathbf{L}_{q+1:,q+1:}\mathbf{L}_{q+1:,q+1:}^\top \quad (16)$$

Here, we use $\mathbf{X}^\top$ since $\mathbf{X}_{:,q:}^\top\mathbf{H}_{-q:}^{-1} = \mathbf{X}^\top\mathbf{H}_{-q:}^{-1}$. To optimize the computation of $\mathbf{P}$, we begin by computing the

---

**Algorithm 1** GPTAQ quantization for one layer

**Input:** FP weight $\mathbf{W}$, calibration input $\mathbf{X}$, FP input $\tilde{\mathbf{X}}$, and Block size $B$
$\mathbf{H} \leftarrow \mathbf{X}\mathbf{X}^\top, \Delta\mathbf{X}\mathbf{X}^\top \leftarrow (\tilde{\mathbf{X}} - \mathbf{X})\mathbf{X}^\top$
$\mathbf{L} = Inverse\_Cholesky(\mathbf{H} + \lambda_1\mathbf{I})$
$\mathbf{P} \leftarrow \Big((\Delta\mathbf{X}\mathbf{X}^\top\mathbf{L}) \odot \mathbf{M_U}\Big)\mathbf{L}^\top$
$\mathbf{Q} \leftarrow \mathbf{0}_{m\times n}, \mathbf{E} \leftarrow \mathbf{0}_{m\times B}$
**for** $i = 0, B, 2B, \dots$ **do**
    **for** $j = i, i+1, \dots, i+B-1$ **do**
      $\mathbf{Q}_{:,j} \leftarrow \text{quant}(\mathbf{W}_{:,j})$
      $\mathbf{E}_{:,j-i} \leftarrow (\mathbf{W}_{:,j} - \mathbf{Q}_{:,j})/\mathbf{L}_{jj}$
      $\mathbf{W}_{:,j:(i+B)} \leftarrow \mathbf{W}_{:,j:(i+B)} - \mathbf{E}_{:,j-i}\mathbf{L}_{j,j:(i+B)}^\top$
                  $+ \mathbf{W}_{:,j}\mathbf{P}_{j,j:(i+B)}$
    **end for**
    $\mathbf{W}_{:,(i+B):} \leftarrow \mathbf{W}_{:,(i+B):} - \mathbf{E}\cdot\mathbf{L}_{i:(i+B),(i+B):}^\top$
                $+ \mathbf{W}_{:,i:(i+B)}\mathbf{P}_{i:(i+B),(i+B):}$
**end for**

---

full matrix $\Delta\mathbf{X}\mathbf{X}^\top \in \mathbb{R}^{n\times n}$. The key challenge becomes efficiently utilizing the structure of $\mathbf{L}_{q+1:,q+1:}\mathbf{L}_{q+1:,q+1:}^\top$ to enable parallel processing of each row in $\mathbf{P}$ on GPUs. The lower-triangular structure of $\mathbf{L}_{q+1:,q+1:}$ presents an opportunity for computational optimization through vectorization, as formalized in the following theorem:

**Theorem 4.2.** *The matrix $\mathbf{P}$ is equal to*

$$\mathbf{P} = \Big((\Delta\mathbf{X}\mathbf{X}^\top\mathbf{L}) \odot \mathbf{M_U}\Big)\mathbf{L}^\top, \quad (17)$$

*where $\mathbf{M_U} \in {0, 1}^{n\times n}$ is a strictly upper-triangular masking matrix with ones above the diagonal and $\odot$ denotes element-wise multiplication.*

Proof is provided in Appendix A.3. With the above formula, we can compute $\mathbf{P}$ in one line code (See Fig. 3 bottom). Furthermore, the proposed method integrates seamlessly with GPTQ's existing implementation, as it also utilizes the Cholesky factor $\mathbf{L}$ for computing diagonal values and the $q$-th row vector $\tilde{\mathbf{H}}_{q,:}^{-1}$. Our implementation extends the original GPTQ framework with minimal modifications.

**Step 4, Lazy-Batch Updates.** With matrix $\mathbf{P}$ computed in advance, we can easily obtain the second term. Moreover, just like GPTQ, the quantization results for column $q$ are only affected by updates performed on this very column, and so updates to later columns are irrelevant at this point in the process. Thus we can *lazily batch* both terms to achieve better GPU utilization. Concretely, given a block of columns $Q$, we first update the columns in the block with a sliced $\mathbf{P}$ and $\mathbf{L}$, and then update the remaining weights outside the block with

$$\Delta\mathbf{W}_{:,Q:} = \frac{(\hat{\mathbf{W}}_{:,Q} - \mathbf{W}_{:,Q})}{\mathbf{L}_{QQ}^\top}\mathbf{L}_{Q,Q:}^\top + \mathbf{W}_{:,Q}\mathbf{P}_{Q,Q:} \quad (18)$$

*Table 1.* Quantization results of vision transformer (left) and language transformer (right). For vision models, we report ImageNet accuracy ($\uparrow$) and for language models, we report Wikitext2 perplexity ($\downarrow$). * denotes our implementation.

| Precision | Method | FT-Free | DeiT-S | DeiT-B | Method | FT-Free | L3-8B | L3-70B | L2-7B | L2-13B | L2-70B |
|---|---|---|---|---|---|---|---|---|---|---|---|
| FP16 | Pretrained | - | 79.8 | 81.3 | Pretrained | - | 6.14 | 2.85 | 5.47 | 4.88 | 3.32 |
| W4A4 | PTQ4ViT | ✓ | 34.1 | 64.4 | OmniQuant | ✗ | - | - | 14.3 | 12.3 | 41.1 |
| | APQ-ViT | ✓ | 43.6 | 67.5 | QLLM | ✗ | - | - | 11.8 | 9.09 | 7.00 |
| | PD-Quant | ✗ | 64.9 | 60.1 | DuQuant | ✗ | 8.06 | - | 6.08 | 5.33 | 3.76 |
| | RepQ-ViT | ✓ | 69.0 | 75.6 | QuaRot | ✓ | 9.91 | 46.6 | 7.98 | 5.93 | 4.03 |
| | GPTQ | ✓ | 71.9 | 77.7 | QuaRot+GPTQ | ✓ | 7.80 | 9.44 | 6.00 | 5.30 | 3.71 |
| | GPTAQ | ✓ | **72.8** | **78.4** | QuaRot+ GPTAQ | ✓ | **7.37** | **6.93** | **5.86** | **5.17** | **3.69** |
| W2A4 | RepQ-ViT* | ✓ | 0.23 | 0.30 | QuaRot | ✓ | 6.0e5 | 3.9e4 | 7.7e3 | 5.6e3 | 1.5e3 |
| | GPTQ | ✓ | 38.4 | 61.0 | QuaRot+GPTQ | ✓ | 102 | 444 | 32.6 | 15.5 | 6.68 |
| | GPTAQ | ✓ | **46.8** | **62.2** | QuaRot+GPTAQ | ✓ | **17.9** | **39.1** | **11.2** | **8.57** | **5.64** |

**Comparing GPTQ with GPTAQ.** Together, we summarized our algorithm in Algorithm 1. The primary additions to GPTQ are **highlighted in blue**. The remaining implementation follows the standard GPTQ procedure, for example, the Hessian diagonal is dampened by 1% of the average values for better numerical stability. It is worthwhile to note that our method is not a simple extension of GPTQ. The second term accounting for residual output error is as important as the quantization error in the first term, as we will show in experiments (subsubsection 5.5.1). Our optimization through neuron decomposition and Cholesky Reformulation ensures both terms can be handled under a unified framework.

## 5. Experiments

### 5.1. Setup

We implement GPTAQ using Hugging Face (Wolf, 2019) on top of the PyTorch framework (Paszke et al., 2019). Unless specifically mentioned, we always use per-channel asymmetric quantization for weights and per-token asymmetric quantization for input activations. The input activation has a clipping ratio of 0.9 as suggested in Ashkboos et al. (2024) and the weight clipping range is searched by minimizing mean squared error (Frantar et al., 2022). We select 128 input samples as calibration dataset, see detailed source in each model type section. For GPTQ implementation, we first quantize weights and then quantize activation following prior work (Ashkboos et al., 2024; Liu et al., 2024), while our GPTAQ quantizes activations in the first place and minimizes layer output residual error in weight quantization[2].

### 5.2. Results on Vision Transformer

We conduct our experiments on DeiT-S/B models (Touvron et al., 2021). We select 128 samples from ImageNet training dataset as calibration data. The compared baselines are PTQ4ViT (Liu et al., 2021), APQ-ViT (Ding et al., 2022), PD-Quant (Liu et al., 2023b), RepQ-ViT (Li et al., 2023), and GPTQ (Frantar et al., 2022). Most of then are

---

[2]A detailed study of quantization order is in Sec. 5.5.2 .

finetuneing-free approaches. On vision transformers, we use *act_order*, an option in GPTQ that sorts the columns based on Hessian diagonal magnitude, which we found useful to improve the performance. The dampening ratio was set to 10% for improved generalization. We test with W2A4 and W4A4 quantization.

We provide the results in Table 1 left part, from which we observe that GPTQ and our GPTAQ outperform the existing quantization regime due to explicit optimization of weights accounting for quantization error minimization. Furthermore, our proposed method outperforms GPTQ baseline by 1% on 4-bit DeiT-S and 0.7% on 4-bit Deit-B model. On W2A4, we run the existing baseline RepQ-ViT which has the best accuracy in W4A4, however, this method fails at this precision (0.23% accuracy for DeiT-S). GPTQ obtains 38.4% accuracy on DeiT-S and our method further improves it to 46.8%.

### 5.3. Results on Language Transformer

We also verify our experiments on large language transformer architectures, including LLaMA2 and LLaMA3. We perform quantization in W4A4 and W2A4 scenarios as we did on the vision transformer. The language transformer is more challenging due to the existence of outliers. Therefore, we apply the incoherence processing with QuaRot (Ashkboos et al., 2024), which is a finetuning-free transformation, to RTN, GPTQ and GPTAQ. We additionally compare several finetuning-based approaches including OmniQuant (Shao et al., 2023), QLLM (Liu et al., 2023a), DuQuant (Lin et al., 2024). Following standard setups, we select 128 2048-token training sequences from the Wikitext2 training set as calibration dataset.

**Perplexity Evaluation.** We first compare the perplexity performance in Table 1 right part. We note that on LLaMA2 models, GPTQ on a rotated model can achieve better performance than other existing methods. Our GPTAQ further improves the perplexity performance. On LLaMA2-7B, our method improves the perplexity from 6.0 to 5.85. For LLaMA3 models, quantization becomes more challenging.

*Table 2.* 4-bit quantization results of LLaMA2/3 Models. We report GPU hours to run quantization (including processing time with rotation, which is not modified by our method), Wikitext2 perplexity and zero-shot reasoning tasks performance.

| Model | Method | FT-Free | GPU Hours | Wiki2(↓) | PiQA | Arc E | Arc C | HellaSwag | Winogrande | BoolQ | Avg(↑) |
|---|---|---|---|---|---|---|---|---|---|---|---|
| | FP16 | ✓ | - | 6.44 | 80.7 | 77.7 | 53.7 | 79.1 | 73.2 | 81.1 | 74.3 |
| | QuaRot+GPTQ | ✓ | 0+0.2 | 7.80 | 75.0 | 70.5 | 43.5 | 73.9 | 66.3 | 73.2 | 67.1 |
| L3-8B | QuaRot+GPTAQ | ✓ | 0+0.3 | **7.36** | **78.2** | **72.7** | **44.8** | **75.4** | **69.1** | **77.5** | **69.6** |
| | SpinQuant+GPTQ | ✗ | 4.0+0.2 | 7.26 | 78.4 | 74.6 | 46.8 | 76.0 | 69.7 | 73.4 | 69.8 |
| | SpinQuant+GPTAQ | ✗ | 4.0+0.3 | **7.19** | **78.7** | **75.9** | **48.4** | **76.1** | **70.6** | **78.2** | **71.3** |
| | FP16 | ✓ | - | 3.32 | 84.4 | 85.9 | 64.3 | 84.9 | 80.7 | 85.1 | 80.4 |
| | QuaRot+GPTQ | ✓ | 0.1+1.8 | 9.44 | 74.6 | 65.0 | 38.7 | 64.2 | 62.7 | 69.3 | 62.4 |
| L3-70B | QuaRot+GPTAQ | ✓ | 0.1+2.7 | **6.94** | **78.2** | **73.2** | **49.0** | **72.6** | **68.3** | **73.3** | **69.1** |
| | SpinQuant+GPTQ | ✗ | 28+1.8 | 6.14 | 79.8 | 76.6 | 54.3 | 78.8 | 73.5 | 80.7 | 73.9 |
| | SpinQuant+GPTAQ | ✗ | 28+2.7 | **5.00** | **82.1** | **81.4** | **58.2** | **82.5** | **76.8** | **85.1** | **77.7** |
| | FP16 | ✓ | - | 5.47 | 79.0 | 74.6 | 46.5 | 76.0 | 68.9 | 77.7 | 70.5 |
| | OmniQuant | ✗ | 2.1 | 14.6 | 65.9 | 43.9 | 30.8 | 53.5 | 55.1 | - | 49.9 |
| | QLLM | ✗ | 1.1 | 11.8 | 67.7 | 44.4 | 30.9 | 58.5 | 56.6 | - | 51.6 |
| L2-7B | DuQuant | ✗ | 2.1 | 6.08 | 75.7 | 50.0 | 37.5 | 69.7 | 63.9 | 69.2 | 61.0 |
| | QuaRot+GPTQ | ✓ | 0+0.2 | 6.00 | 77.2 | 70.4 | 42.6 | 73.0 | 65.7 | 74.5 | 67.2 |
| | QuaRot+GPTAQ | ✓ | 0+0.3 | **5.85** | **77.3** | **70.6** | **43.7** | **73.6** | **67.6** | **75.8** | **68.1** |
| | SpinQuant+GPTQ | ✗ | 3.3+0.2 | 5.90 | 76.7 | 70.2 | 43.3 | 73.1 | 65.2 | 73.7 | 67.1 |
| | SpinQuant+GPTAQ | ✗ | 3.3+0.3 | **5.85** | **77.3** | **71.9** | **43.8** | 72.9 | **68.0** | **74.3** | **68.0** |
| | FP16 | ✓ | - | 4.88 | 80.5 | 77.5 | 49.2 | 79.4 | 72.4 | 80.6 | 73.3 |
| | OmniQuant | ✗ | 3.2 | 12.3 | 69.8 | 47.2 | 33.8 | 59.3 | 55.5 | - | 53.1 |
| | QLLM | ✗ | 1.7 | 9.09 | 70.5 | 48.5 | 34.4 | 62.8 | 55.4 | - | 54.3 |
| L2-13B | DuQuant | ✗ | 3.2 | 5.33 | 77.3 | 56.2 | 42.2 | 73.7 | 65.4 | 63.4 | 63.8 |
| | QuaRot+GPTQ | ✓ | 0+0.4 | 5.30 | 78.1 | 74.2 | 46.4 | 76.5 | **70.4** | 78.3 | 70.6 |
| | QuaRot+GPTAQ | ✓ | 0+0.5 | **5.17** | **78.7** | **74.8** | **47.4** | **77.5** | 70.1 | **78.8** | **71.2** |
| | SpinQuant+GPTQ | ✗ | 4.0+0.4 | 5.20 | 78.2 | 75.3 | 47.3 | 76.9 | 67.7 | 77.1 | 70.7 |
| | SpinQuant+GPTAQ | ✗ | 4.0+0.5 | **5.19** | **78.9** | **75.6** | **48.8** | **77.3** | **71.0** | **77.7** | **71.5** |
| | FP16 | ✓ | - | 3.32 | 82.7 | 81.0 | 57.3 | 83.8 | 78.0 | 83.8 | 77.8 |
| | OmniQuant | ✗ | 15 | 41.1 | 53.0 | 31.2 | 23.9 | 33.9 | 52.0 | - | 38.8 |
| | QLLM | ✗ | 9.3 | 7.00 | 74.3 | 50.6 | 37.2 | 71.6 | 59.4 | - | 58.6 |
| L2-70B | DuQuant | ✗ | 15 | 3.76 | 79.8 | 59.8 | 46.8 | 79.4 | 74.1 | 73.1 | 68.8 |
| | QuaRot+GPTQ | ✓ | 0.1+1.8 | 3.71 | 81.6 | **79.7** | 55.6 | 81.8 | 76.6 | 81.5 | 76.1 |
| | QuaRot+GPTAQ | ✓ | 0.1+2.7 | **3.69** | **82.4** | 79.0 | **55.9** | 82.1 | **76.8** | 82.3 | **76.4** |
| | SpinQuant+GPTQ | ✗ | 28+1.8 | 3.68 | **82.0** | 79.0 | **55.6** | 82.3 | **76.5** | 82.6 | 76.3 |
| | SpinQuant+GPTAQ | ✗ | 28+2.7 | **3.67** | **82.0** | **79.3** | 55.4 | **82.4** | 76.4 | **82.7** | **76.4** |

For example, GPTQ in 4-bit quantization increased the perplexity of LLaMA3-70B to 9.44, while our method reduced this to 6.93. To the best of our knowledge, we are the first to validate the W2A4 perplexity results in language transformers. In this case, the quantized model easily crashed without a delicate calibration, for example, applying QuaRot and RTN consistently yields > 1000 perplexity, significantly deteriorating the model performance. Applying GPTQ can recover the model but still degrades the full-precision performance by a large margin. Remarkably, our method obtains great improvement over GPTQ, reducing the perplexity of GPTQ by 20%∼90%.

**Zero-Shot Accuracy Evaluation.** We evaluate the quantized model's zero-shot performance on six downstream tasks: PiQA (Bisk et al., 2020), ARC easy / challenged (Clark et al., 2018), Hellaswag (Zellers et al., 2019), Winogrande (Sakaguchi et al., 2021), and BoolQ (Clark et al., 2019). In addition to QuaRot, we take the learned rotation matrices from SpinQuant (Liu et al., 2024), a finetuning-based approach, to combine it with our GPTAQ. To ensure a comprehensive comparison in effectiveness and efficiency, we additionally report the GPU Hours (on one A100) re-

quired to run the algorithm. Although in practice, Spin-Quants runs on 8 A100 GPUs. The results are shown in Table 2. With QuaRot, GPTQ has 7.2% and 18% gap with the full precision 8B and 70B models, respectively. GP-TAQ can effectively reduce these gaps to 4.7% and 11%. Notably, QuaRot+GPTAQ can achieve similar average accuracy with SpinQuant+GPTQ, an approach that takes significantly more time than finetuning-free quantization. For example, on LLaMA2-7B, QuaRot+GPTAQ achieves 5.85 perplexity and 68.1% average accuracy, which is even 0.05 lower and 1% higher than the SpinQuant+GPTQ.

Since SpinQuant primarily focuses on activation quantization and our method can incorporate this part into our asymmetric calibration, we further test SpinQuant+GPTAQ using the official checkpoints. GPTAQ can improve the performance of SpinQuant as well. For instance, on LLaMA3-70B, our method improves the average accuracy from 73.9% to 77.7% (3.8% improvement).

**Results on Weight-Only Quantization.** Given that the original GPTQ method mainly focuses on the weight-only quantization, we also evaluate our method under this case. To make a fair comparison, we use the original paper's

*Table 3.* 3-bit per-group weight symmetric quantization results of LLaMA2/3 Models. We report perplexity on Wikitext2 and C4 and the reasoning accuracy on 8 datasets. We use 128 examples from the C4 datasets (Raffel et al., 2020) to calibrate the model.

| Model | Method | Wiki2(↓) | C4(↓) | PiQA | Arc E | Arc C | HS | WG | BoolQ | OBQA | SiQA | Avg(↑) |
|---|---|---|---|---|---|---|---|---|---|---|---|---|
| L3-8B-Instruct | FP16 | 7.21 | 11.39 | 81.2 | 79.3 | 55.0 | 79.2 | 73.7 | 84.1 | 43.2 | 32.9 | 66.1 |
| | AWQ | 10.47 | 16.77 | 76.7 | 71.1 | 46.3 | 72.5 | 70.3 | 80.9 | 39.8 | 32.9 | 61.3 |
| | GPTQ | 9.04 | 14.01 | 78.1 | 72.0 | 48.0 | 74.0 | 72.5 | 81.4 | 41.6 | 32.8 | 62.5 |
| | GPTAQ | 8.86 | 13.84 | 78.7 | 77.2 | 50.6 | 74.9 | 71.6 | 83.2 | 41.6 | 32.9 | **63.8** |
| L2-7B | FP16 | 5.47 | 7.26 | 79.1 | 74.5 | 46.3 | 76.0 | 69.0 | 77.7 | 44.2 | 32.9 | 62.5 |
| | AWQ | 6.75 | 8.98 | 76.4 | 67.2 | 42.1 | 71.8 | 67.6 | 69.1 | 40.6 | 33.8 | 58.6 |
| | GPTQ | 6.88 | 14.02 | 76.6 | 65.0 | 38.5 | 67.5 | 67.6 | 72.4 | 41.2 | 33.5 | 57.8 |
| | GPTAQ | 6.59 | 8.39 | 78.0 | 68.6 | 42.2 | 72.9 | 65.7 | 72.2 | 40.2 | 33.1 | **59.1** |
| L2-13B | FP16 | 4.88 | 6.72 | 80.5 | 77.5 | 49.2 | 79.4 | 72.2 | 80.6 | 45.2 | 33.2 | 64.7 |
| | AWQ | 5.49 | 7.57 | 78.6 | 74.4 | 43.1 | 76.2 | 71.9 | 77.7 | 44.4 | 32.8 | 62.4 |
| | GPTQ | 5.42 | 7.36 | 79.7 | 75.0 | 47.3 | 75.6 | 71.4 | 79.6 | 43.8 | 32.9 | 63.1 |
| | GPTAQ | 5.41 | 7.33 | 79.6 | 76.1 | 48.5 | 76.3 | 72.0 | 81.8 | 43.2 | 33.1 | **63.8** |

*Table 4.* Quantization results of huge transformers.

| Precision | Method | FT-Free | EVA-02 | L3.1-405B |
|---|---|---|---|---|
| FP16 | Pretrained | - | 90.05 (↑) | 1.44 (↓) |
| W4A4 | RTN | ✓ | 85.72 | 17.4 |
| | GPTQ | ✓ | 86.48 | 5.82 |
| | GPTAQ | ✓ | **88.30** | **3.48** |

*Table 5.* Ablation study of $\Delta\mathbf{W}$ on LLaMA3-8B.

| Precision | Method | $\Delta\mathbf{W}$ | Wiki2(↓) | Avg(↑) |
|---|---|---|---|---|
| W4A4 | RTN | $\mathbf{0}$ | 9.91 | 65.5 |
| | GPTQ | $\mathbf{E}_{:,q}\mathbf{L}_{q,:}^\top$ | 7.80 | 67.1 |
| | GPTAQ′ | $\mathbf{W}_{:,q}\mathbf{P}_{q,:}$ | 7.97 | 69.0 |
| | GPTAQ | $\mathbf{E}_{:,q}\mathbf{L}_{q,:}^\top + \mathbf{W}_{:,q}\mathbf{P}_{q,:}$ | **7.36** | **69.6** |

primary settings: 3-bit per-group weight quantization. We use a symmetric format (no zero point), and the group size is set to 128. We turn on *act order* operation that sorts the Hessian based on its diagonal values. Additionally, we test another weight-only quantization algorithm, AWQ (Lin et al., 2023), for its layer-wise optimization. As shown in Table 3, GPTAQ is able to achieve the best performance among these layer-wise FT-free quantization algorithms. Remarkably, for the LLaMA3-8B-Instruct, we have 63.8% average accuracy, which is 2.4% higher than AWQ and 1.3% higher than GPTQ.

### 5.4. Results on Huge Transformers

To demonstrate the scalability of our method, we further test two huge transformers architectures, (1) EVA-02 (Fang et al., 2024), the Rank 1st architecture in Pytorch Image Models benchmark that achieves **90.05%** ImageNet top-1 accuracy, and (2) LLaMA3.1-405B that achieves **1.44** Wikitext2 perplexity. We perform GPTQ and GPTAQ **on a single A100 GPU** to quantize these two models into 4-bit weights and activations. The results are shown in Table 4. For EVA-02, the RTN and GPTQ method degrades the full precision accuracy by 4.3% and 3.5% respectively. Our GPTAQ, notably, reduces this gap to 1.7%, which is half of the GPTQ algorithm. For LLaMA3.1-405B, which contains 126 transformer blocks with an intermediate size of 8096, the RTN significantly degrades its perplexity performance to 17.4. Our GPTAQ achieves 4.32 perplexity. Compared to the GPTQ algorithm, our method effectively reduces perplexity by 2.3, demonstrating the scalability of our method.

### 5.5. Ablation Study

#### 5.5.1. WEIGHT UPDATE

The proposed algorithm comprises two different terms $\mathbf{E}_{:,q}\mathbf{L}_{q,:}^\top$ and $\mathbf{W}_{:,q}\mathbf{P}_{q,:}$ for $\Delta\mathbf{W}$, which can be viewed as minimizing the quantization error from current layer, and minimizing the quantization error from the previous layer, respectively. Therefore, we test the performance of applying these two terms *individually* and observe how they contribute to the final performance. We conduct experiments on W4A4 LLaMA3-8B with QuaRot transformations. The results are demonstrated in Table 5. Note that if we do not empower any update to weights, the method will reduce to RTN; and GPTQ is the case where we only apply the first term. Interestingly, we can find that solely applying the first term or second term can increase the quantization performance compared to RTN. While applying the first term (GPTQ) obtains a better perplexity score, the average accuracy when applying the second term individually is much better, resulting in 2% higher performance. Combining both terms, which is our GPTAQ, the quantization performance can be further improved. This result suggests that quantization errors accumulated in previous layers should be taken into account during calibration.

#### 5.5.2. ACTIVATION QUANTIZATION ORDER

Conventionally, for GPTQ activation quantization is added after weight quantization is done as did in QuaRot (Ashkboos et al., 2024) and SpinQuant (Liu et al., 2024). For our GPTAQ, the activation quantization, however, is added

*Table 6.* Comparison of activation/weight quantization pipeline. All weights and activations are quantized into 4-bit.

| Model | Method | Q Order | Wiki2($\downarrow$) | PiQA | Arc E | Arc C | HellaSwag | Winogrande | BoolQ | Avg($\uparrow$) |
|---|---|---|---|---|---|---|---|---|---|---|
| | FP16 | - | 6.44 | 80.7 | 77.7 | 53.7 | 79.1 | 73.2 | 81.1 | 74.3 |
| | QuaRot+GPTQ | W$\rightarrow$A | 7.80 | 75.0 | 70.5 | 43.5 | 73.9 | 66.3 | 73.2 | 67.1 |
| L3-8B | QuaRot + GPTAQ | W$\rightarrow$A | 7.68 | 78.2 | 73.8 | 45.8 | 74.3 | 68.4 | 74.6 | 69.2 |
| | QuaRot+GPTQ | A$\rightarrow$W | 7.78 | 76.3 | 72.3 | 45.3 | 73.8 | 67.7 | 75.8 | 68.6 |
| | QuaRot + GPTAQ | A$\rightarrow$W | **7.36** | **78.2** | **72.7** | **44.8** | **75.4** | **69.1** | **77.5** | **69.6** |
| | FP16 | - | 5.47 | 79.0 | 74.6 | 46.5 | 76.0 | 68.9 | 77.7 | 70.5 |
| | QuaRot+GPTQ | W$\rightarrow$A | 6.00 | 77.2 | 70.4 | 42.6 | 73.0 | 65.7 | 74.5 | 67.2 |
| L2-7B | QuaRot + GPTAQ | W$\rightarrow$A | 5.95 | 77.3 | 70.6 | 42.5 | 73.7 | 66.5 | 75.5 | 67.7 |
| | QuaRot+GPTQ | A$\rightarrow$W | 6.00 | 76.7 | 72.0 | 42.4 | 73.1 | 65.8 | 75.1 | 67.5 |
| | QuaRot + GPTAQ | A$\rightarrow$W | **5.85** | **77.3** | **70.6** | **43.7** | **73.6** | **67.6** | **75.8** | **68.1** |

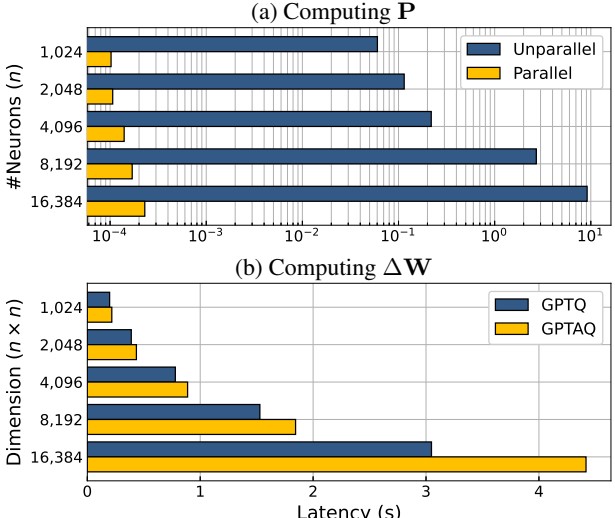

*Figure 4.* Latency visualization of our algorithm under various $n$.

before weight quantization so that $\Delta\mathbf{X}$ can capture this information. We now demonstrate the cases where activation quantization is added before GPTQ or after GPTQ.

We conduct experiments on 4-bit LLaMA3-8B and LLaMA2-7B. The results are shown in Table 6. From the table, we can conclude that (1) For GPTQ calibration whether the activation quantization is enabled or not has no impact on the perplexity results. (2) However, activation quantization may help GPTQ achieve higher downstream task accuracy. (3) For GPTAQ, enabling activation quantization can further aid the performance as it considers more information in input activation error. (4) Even if we disable the activation quantization, GPTAQ still outperforms GPTQ regardless of quantization order.

### 5.6. Algorithm Efficiency

We analyze the speed of running GPTAQ. (Memory analysis is presented in Appendix C). First, we compare the latency of calculating $\mathbf{P}$ using unparalleled implementation (Eq. (16)) and paralleled implementation (Eq. (17)) under various size of $n$. Second, we compare the latency of GPTQ and GPTAQ on a layer with $n \times n$ dimension, using the pro-

cedure in Algorithm 1. We assume that $\Delta\mathbf{X}\mathbf{X}^\top$ and $\mathbf{L}$ are obtained previously, and test the latency on one A100 GPU with PyTorch 2.4.1-cu12.4. In Fig. 4(a), we first compare the latency to compute $\mathbf{P}$. Thanks to the highly optimized CUDA kernel, our parallel implementation takes less than $1ms$ to finish, which is $> 10^4$ faster than the unparalleled implementation despite using the same number of operations. Fig. 4(b) compares the speed of running GPTQ and GPTAQ, from which we observe that GPTAQ incurs less than 10% more latency when weight dimension is smaller than 4096. In these cases, the latency bottleneck is the quantization operation rather than computing $\Delta\mathbf{W}$. When dimension further expands, the bottleneck switches to computing $\Delta\mathbf{W}$, and our GPTAQ increases the latency by a slight margin (30$\sim$40% latency).

## Conclusion

In this paper, we introduce GPTAQ, an efficient finetuning-free quantization to reduce the accumulated asymmetry error in quantization. Building upon the OBQ framework, our method introduces 4 steps that effectively parallelize and accelerate the computation of optimal weight update. As a result, GPTAQ is easy to implement and can adapt to the previous GPTQ framework with minimum effort. Through both qualitative and quantitative verification, GPTAQ can effectively reduce the asymmetry to improve the quantization performance without involving finetuning.

## Impact Statement

This paper presents work whose goal is to advance the field of model compression. There are many potential societal consequences of our work, none which we feel must be specifically highlighted here.

## Acknowledgment

This work was supported in part by CoCoSys, a JUMP2.0 center sponsored by DARPA and SRC, the National Science Foundation (CAREER Award, Grant #2312366, Grant #2318152), and the DoE MMICC center SEA-CROGS (Award #DE-SC0023198).

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

# A. Theoretical Derivation

## A.1. Optimal Framework

We provide a detailed derivation of the optimal framework. To compute the local minima of the Lagrangian form, we set the partial derivatives to zeros:

$$
\begin{cases}
\dfrac{\partial L}{\partial \Delta \mathbf{w}} = 2\Delta \mathbf{w}\mathbf{H} - 2\mathbf{r}\mathbf{X}^\top + \lambda \mathbf{e}_q = 0 \\[2mm]
\dfrac{\partial L}{\partial \lambda} = \Delta \mathbf{w}\mathbf{e}_q^\top + \mathbf{w}_q - \hat{\mathbf{w}}_q = 0
\end{cases}
\tag{19}
$$

Solving the first equation, we have

$$
\Delta \mathbf{w}\mathbf{H} = -\frac{\lambda}{2}\mathbf{e}_q + \mathbf{r}\mathbf{X}^\top.
\tag{20}
$$

Right multiplying the inverse Hessian on both sides, we get

$$
\Delta \mathbf{w} = -\frac{\lambda}{2}\mathbf{e}_q\mathbf{H}^{-1} + \mathbf{r}\mathbf{X}^\top\mathbf{H}^{-1}
\tag{21a}
$$

$$
= -\frac{\lambda}{2}\mathbf{H}_{q,:}^{-1} + \mathbf{r}\mathbf{X}^\top\mathbf{H}^{-1}
\tag{21b}
$$

Substituting the equation into the second equation of Eq. (A.1), we have

$$
-\frac{\lambda}{2}\mathbf{H}_{q,:}^{-1}\mathbf{e}_q^\top + \mathbf{r}\mathbf{X}^\top\mathbf{H}^{-1}\mathbf{e}_q^\top + \mathbf{w}_q - \hat{\mathbf{w}}_q = -\frac{\lambda}{2}\mathbf{H}_{qq}^{-1} + \mathbf{r}\mathbf{X}^\top\mathbf{H}_{:,q}^{-1} + \mathbf{w}_q - \hat{\mathbf{w}}_q = 0
\tag{22a}
$$

where we can easily compute $\lambda$ as

$$
\lambda = 2\left( \frac{\mathbf{w}_q - \hat{\mathbf{w}}_q}{\mathbf{H}_{qq}^{-1}} + \frac{\mathbf{r}\mathbf{X}^\top\mathbf{H}_{:,q}^{-1}}{\mathbf{H}_{qq}^{-1}} \right)
\tag{23}
$$

Now substituting the above equation into Eq. (21b), we can compute $\Delta \mathbf{w}$ as

$$
\Delta \mathbf{w} = -\frac{\lambda}{2}\mathbf{H}_{q,:}^{-1} + \mathbf{r}\mathbf{X}^\top\mathbf{H}^{-1}
\tag{24a}
$$

$$
= -\frac{\mathbf{w}_q - \hat{\mathbf{w}}_q}{\mathbf{H}_{qq}^{-1}} \cdot (\mathbf{H}_{q,:}^{-1}) - \frac{\mathbf{r}\mathbf{X}^\top\mathbf{H}_{:,q}^{-1}\mathbf{H}_{q,:}^{-1}}{\mathbf{H}_{qq}^{-1}} + \mathbf{r}\mathbf{X}^\top\mathbf{H}^{-1}
\tag{24b}
$$

$$
= -\frac{\mathbf{w}_q - \hat{\mathbf{w}}_q}{\mathbf{H}_{qq}^{-1}} \cdot (\mathbf{H}_{q,:}^{-1}) + \mathbf{r}\mathbf{X}^\top \left( \mathbf{H}^{-1} - \frac{\mathbf{H}_{:,q}^{-1}\mathbf{H}_{q,:}^{-1}}{\mathbf{H}_{qq}^{-1}} \right)
\tag{24c}
$$

$$
= -\frac{\mathbf{w}_q - \hat{\mathbf{w}}_q}{\mathbf{H}_{qq}^{-1}} \cdot (\mathbf{H}_{q,:}^{-1}) + \mathbf{r}\mathbf{X}^\top\mathbf{H}_{-q}^{-1}
\tag{24d}
$$

The $L_q$ is derived by replacing the optimal $\Delta \mathbf{w}$ into the loss function $||\Delta \mathbf{w}\mathbf{X} - \mathbf{r}||_F^2$, given by

$$
L_q = ||\mathbf{r}\mathbf{X}^\top\mathbf{H}_{-q}^{-1}\mathbf{X} - \mathbf{r} + \frac{(\hat{\mathbf{w}}_q - \mathbf{w}_q)}{\mathbf{H}_{qq}^{-1}} \cdot (\mathbf{H}_{q,:}^{-1})\mathbf{X}||_F^2.
\tag{25}
$$

Expanding this function, we obtain

$$
L_q = \mathbf{r}\mathbf{X}^\top\mathbf{H}_{-q}^{-1}\mathbf{X}\mathbf{X}^\top\mathbf{H}_{-q}^{-1}\mathbf{X}\mathbf{r}^\top + \mathbf{r}\mathbf{r}^\top - 2\mathbf{r}\mathbf{X}^\top\mathbf{H}_{-q}^{-1}\mathbf{X}\mathbf{r}^\top + \frac{(\hat{\mathbf{w}}_q - \mathbf{w}_q)^2}{(\mathbf{H}_{qq}^{-1})^2} \cdot \mathbf{H}_{q,:}^{-1}\mathbf{X}\mathbf{X}^\top\mathbf{H}_{:,q}^{-1}
\tag{26a}
$$

$$
+ 2\frac{(\hat{\mathbf{w}}_q - \mathbf{w}_q)}{\mathbf{H}_{qq}^{-1}} \cdot \left(\mathbf{r}\mathbf{X}^\top\mathbf{H}_{-q}^{-1}\mathbf{X}\mathbf{X}^\top\mathbf{H}_{:,q}^{-1}\right) - 2\frac{(\hat{\mathbf{w}}_q - \mathbf{w}_q)}{\mathbf{H}_{qq}^{-1}} \cdot \left(\mathbf{r}\mathbf{X}^\top\mathbf{H}_{:,q}^{-1}\right)
\tag{26b}
$$

$$
= \mathbf{r}\mathbf{X}^\top\mathbf{H}_{-q}^{-1}\mathbf{H}\mathbf{H}_{-q}^{-1}\mathbf{X}\mathbf{r}^\top + \mathbf{r}\mathbf{r}^\top - 2\mathbf{r}\mathbf{X}^\top\mathbf{H}_{-q}^{-1}\mathbf{X}\mathbf{r}^\top + \frac{(\hat{\mathbf{w}}_q - \mathbf{w}_q)^2}{(\mathbf{H}_{qq}^{-1})^2} \cdot \mathbf{H}_{q,:}^{-1}\mathbf{H}\mathbf{H}_{:,q}^{-1}
\tag{26c}
$$

$$
+ 2\frac{(\hat{\mathbf{w}}_q - \mathbf{w}_q)}{\mathbf{H}_{qq}^{-1}} \cdot \left(\mathbf{r}\mathbf{X}^\top\mathbf{H}_{-q}^{-1}\mathbf{H}\mathbf{H}_{:,q}^{-1}\right) - 2\frac{(\hat{\mathbf{w}}_q - \mathbf{w}_q)}{\mathbf{H}_{qq}^{-1}} \cdot \left(\mathbf{r}\mathbf{X}^\top\mathbf{H}_{:,q}^{-1}\right)
\tag{26d}
$$

We start by simplifying $\mathbf{H}_{-q}^{-1}\mathbf{H}$:

$$\mathbf{H}_{-q}^{-1}\mathbf{H} = \left(\mathbf{H}^{-1} - \frac{\mathbf{H}_{:,q}^{-1}\mathbf{H}_{q,:}^{-1}}{\mathbf{H}_{qq}^{-1}}\right)\mathbf{H} \tag{27a}$$

$$= \mathbf{I} - \left(\frac{\mathbf{H}_{:,q}^{-1}\mathbf{H}_{q,:}^{-1}}{\mathbf{H}_{qq}^{-1}}\mathbf{H}\right) \tag{27b}$$

$$= \mathbf{I} - \left(\frac{\mathbf{H}^{-1}\mathbf{e}_q^\top\mathbf{e}_q\mathbf{H}^{-1}}{\mathbf{H}_{qq}^{-1}}\mathbf{H}\right) \tag{27c}$$

$$= \mathbf{I} - \frac{1}{\mathbf{H}_{qq}^{-1}}\mathbf{H}^{-1}\mathbf{e}_q^\top\mathbf{e}_q \tag{27d}$$

Substitute this function back into Eq. (26d), we get

$$L_q = \mathbf{r}\mathbf{X}^\top\left(\mathbf{I} - \frac{1}{\mathbf{H}_{qq}^{-1}}\mathbf{H}^{-1}\mathbf{e}_q^\top\mathbf{e}_q\right)\mathbf{H}_{-q}^{-1}\mathbf{X}\mathbf{r}^\top + \mathbf{r}\mathbf{r}^\top - 2\mathbf{r}\mathbf{X}^\top\mathbf{H}_{-q}^{-1}\mathbf{X}\mathbf{r}^\top + \frac{(\hat{\mathbf{w}}_q - \mathbf{w}_q)^2}{(\mathbf{H}_{qq}^{-1})^2}\cdot\mathbf{H}_{q,:}^{-1}\mathbf{H}\mathbf{H}_{:,q}^{-1} \tag{28a}$$

$$+ 2\frac{(\hat{\mathbf{w}}_q - \mathbf{w}_q)}{\mathbf{H}_{qq}^{-1}}\cdot\left(\mathbf{r}\mathbf{X}^\top\left(\mathbf{I} - \frac{1}{\mathbf{H}_{qq}^{-1}}\mathbf{H}^{-1}\mathbf{e}_q^\top\mathbf{e}_q\right)\mathbf{H}_{:,q}^{-1}\right) - 2\frac{(\hat{\mathbf{w}}_q - \mathbf{w}_q)}{\mathbf{H}_{qq}^{-1}}\cdot\left(\mathbf{r}\mathbf{X}^\top\mathbf{H}_{:,q}^{-1}\right) \tag{28b}$$

$$= -\frac{1}{\mathbf{H}_{qq}^{-1}}\mathbf{r}\mathbf{X}^\top\mathbf{H}^{-1}\mathbf{e}_q^\top\mathbf{e}_q\mathbf{H}_{-q}^{-1}\mathbf{X}\mathbf{r}^\top + \mathbf{r}\mathbf{r}^\top - \mathbf{r}\mathbf{X}^\top\mathbf{H}_{-q}^{-1}\mathbf{X}\mathbf{r}^\top + \frac{(\hat{\mathbf{w}}_q - \mathbf{w}_q)^2}{(\mathbf{H}_{qq}^{-1})^2}\cdot\mathbf{H}_{q,:}^{-1}\mathbf{H}\mathbf{H}_{:,q}^{-1} \tag{28c}$$

$$- 2\frac{(\hat{\mathbf{w}}_q - \mathbf{w}_q)}{(\mathbf{H}_{qq}^{-1})^2}\mathbf{r}\mathbf{X}^\top\mathbf{H}^{-1}\mathbf{e}_q^\top\mathbf{e}_q\mathbf{H}_{:,q}^{-1}, \tag{28d}$$

We note that the $\mathbf{e}_q\mathbf{H}_{-q}^{-1} = \mathbf{0}$ is a all-zero vector since the $q$-th row of $\mathbf{H}_{-q}^{-1}$ is eliminated. Therefore, the first term in Eq. (28d) is omitted. For last term, we can rewrite it to $-2\frac{(\hat{\mathbf{w}}_q - \mathbf{w}_q)}{\mathbf{H}_{qq}^{-1}}\mathbf{r}\mathbf{X}^\top\mathbf{H}_{:,q}^{-1}$ due to $\mathbf{e}_q\mathbf{H}_{:,q}^{-1} = \mathbf{H}_{qq}^{-1}$.

We further simplify $\mathbf{H}_{q,:}^{-1}\mathbf{H}\mathbf{H}_{:,q}^{-1}$ by

$$\mathbf{H}_{q,:}^{-1}\mathbf{H}\mathbf{H}_{:,q}^{-1} = \mathbf{e}_q\mathbf{H}^{-1}\mathbf{H}\mathbf{H}^{-1}\mathbf{e}_q^\top \tag{29a}$$

$$= \mathbf{e}_q\mathbf{H}^{-1}\mathbf{e}_q^\top \tag{29b}$$

$$= \mathbf{H}_{qq}^{-1} \tag{29c}$$

To this end, we can simplify the $L_q$ to

$$L_q = \frac{(\hat{\mathbf{w}}_q - \mathbf{w}_q)^2}{\mathbf{H}_{qq}^{-1}} + \mathbf{r}\mathbf{r}^\top - \mathbf{r}\mathbf{X}^\top\mathbf{H}_{-q}^{-1}\mathbf{X}\mathbf{r}^\top - 2\frac{(\hat{\mathbf{w}}_q - \mathbf{w}_q)}{\mathbf{H}_{qq}^{-1}}\mathbf{r}\mathbf{X}^\top\mathbf{H}_{:,q}^{-1} \tag{30}$$

### A.2. Proof of Lemma 4.1

*Proof.* Without loss of generality, we first prove the case of $\mathbf{H}_{-1}^{-1} = (\mathbf{X}_{-1}\mathbf{X}_{-1}^\top)^{-1} = \mathbf{L}_{2:,2:}\mathbf{L}_{2:,2:}^\top$. When performing the Cholesky decomposition on $\mathbf{H}^{-1}$, we have $\mathbf{H}^{-1} = \mathbf{L}\mathbf{L}^\top$ where $\mathbf{L}$ is a lower-triangular matrix. Thus, we can rewrite the Cholesky factor as

$$\mathbf{L} = \begin{bmatrix} \mathbf{L}_{11} & \mathbf{0} \\ \mathbf{L}_{2:,1} & \mathbf{L}_{2:,2:} \end{bmatrix} \tag{31}$$

Now, substitute the $\mathbf{H}^{-1} = \mathbf{L}\mathbf{L}^\top$ with above equation, we have

$$\mathbf{H}^{-1} = \begin{bmatrix} \mathbf{H}_{11}^{-1} & \mathbf{H}_{1,2:}^{-1} \\ \mathbf{H}_{2:,1}^{-1} & \mathbf{H}_{2:,2:}^{-1} \end{bmatrix} = \begin{bmatrix} \mathbf{L}_{11} & \mathbf{0} \\ \mathbf{L}_{2:,1} & \mathbf{L}_{2:,2:} \end{bmatrix}\begin{bmatrix} \mathbf{L}_{11} & \mathbf{L}_{2:,1}^\top \\ \mathbf{0} & \mathbf{L}_{2:,2:}^\top \end{bmatrix} = \begin{bmatrix} \mathbf{L}_{11}^2 & \mathbf{L}_{11}\mathbf{L}_{2:,1}^\top \\ \mathbf{L}_{11}\mathbf{L}_{2:,1} & \mathbf{L}_{2:,1}\mathbf{L}_{2:,1}^\top + \mathbf{L}_{2:,2:}\mathbf{L}_{2:,2:}^\top \end{bmatrix} \tag{32}$$

We can thus construct a linear system from the above equation, given by

$$\begin{cases} \mathbf{H}_{11}^{-1} = \mathbf{L}_{11}^2 \\ \mathbf{H}_{2:,1}^{-1} = \mathbf{L}_{11}\mathbf{L}_{2:,1} \\ \mathbf{H}_{2:,2:}^{-1} = \mathbf{L}_{2:,1}\mathbf{L}_{2:,1}^\top + \mathbf{L}_{2:,2:}\mathbf{L}_{2:,2:}^\top \end{cases} \tag{33}$$

It is straightforward to solve the equations and obtain

$$
\begin{cases}
\mathbf{L}_{11} = \sqrt{\mathbf{H}_{11}^{-1}} \\[2ex]
\mathbf{L}_{2:,1} = \dfrac{1}{\sqrt{\mathbf{H}_{11}^{-1}}} \mathbf{H}_{2:,1}^{-1} \\[2ex]
\mathbf{L}_{2:,2:}\mathbf{L}_{2:,2:}^{\top} = \mathbf{H}_{2:,2:}^{-1} - \dfrac{1}{\sqrt{\mathbf{H}_{11}^{-1}}} \dfrac{1}{\sqrt{\mathbf{H}_{11}^{-1}}} \mathbf{H}_{2:,1}^{-1}\mathbf{H}_{1,2:}^{-1}
\end{cases}
\tag{34}
$$

Recall that $\mathbf{H}_{1:,1:}^{-1} - \frac{1}{\mathbf{H}_{11}^{-1}}\mathbf{H}_{1:,1}^{-1}\mathbf{H}_{1,1:}^{-1}$ performs the Gaussian Elimination on the first row/column, which is the $\mathbf{H}_{-1}^{-1}$. The third equation in Eq. (A.2) is essentially the $\mathbf{H}_{-1}^{-1}$ with the first row/column removed. Therefore, we have $\mathbf{H}_{-1}^{-1} = \mathbf{L}_{2:,2:}\mathbf{L}_{2:,2:}^{\top}$. The lemma can be derived by recursively removing the first row/column in the current inverse Hessian to get $\mathbf{H}_{-q:}^{-1} = \mathbf{L}_{q+1:,q+1:}\mathbf{L}_{q+1:,q+1:}^{\top}$. $\qquad\square$

### A.3. Proof of Theorem 4.2

*Proof.* We start with computing each row of $\mathbf{P}$. Recall that this formula is given by

$$
\mathbf{P}_{i,:} = \Delta\mathbf{X}_{i,:}\mathbf{X}^{\top}\mathbf{L}_{i+1:,i+1:}\mathbf{L}_{i+1:,i+1:}^{\top}
\tag{35}
$$

For matrix $\mathbf{L}_{i+1:,i+1:}^{\top}$, the $j$-th column has non-zero values only if $j > i$. therefore, elements in $\mathbf{P}$ can be written as

$$
\mathbf{P}_{i,j} = \begin{cases} \displaystyle\sum_{a=i+1}^{j} \mathbf{O}_{i,a}\mathbf{L}_{a,j}^{\top} & \text{if } i < j \\[3ex] 0 & \text{if } i \geq j \end{cases}
\tag{36}
$$

where $\mathbf{O}_{i,a} = (\Delta\mathbf{X}_{i,:}\mathbf{X}^{\top}\mathbf{L}_{i+1:,i+1:})_a$ is the element from the product of the first three terms. We again note that $\mathbf{L}_{i+1:,i+1:}$ has non-zero values in $a$-th column only if $a > i$. Thus, the matrix $\mathbf{O}$ is computed as

$$
\mathbf{O}_{i,a} = \begin{cases} \displaystyle\sum_{b=a}^{n}(\Delta\mathbf{X}_{i,:}\mathbf{X}^{\top})_b\mathbf{L}_{b,a} = \sum_{b=a}^{n}(\Delta\mathbf{X}\mathbf{X}^{\top})_{i,b}\mathbf{L}_{b,a} & \text{if } i < a \\[3ex] 0 & \text{if } i \geq a \end{cases}
\tag{37}
$$

Hence, $\mathbf{O}$ can be derived by masking out the lower-triangular area of $\Delta\mathbf{X}\mathbf{X}^{\top}\mathbf{L}$, as

$$
\mathbf{O} = (\Delta\mathbf{X}\mathbf{X}^{\top}\mathbf{L}) \odot \mathbf{M}_{\mathbf{U}}.
\tag{38}
$$

The fact that $\mathbf{O}$ has zeros values on the lower-triangular area makes it possible to directly multiply $\mathbf{O}$ and $\mathbf{L}^{\top}$ to get $\mathbf{P}$, given by

$$
(\mathbf{O}\mathbf{L}^{\top})_{i,j} = \sum_{a=1}^{j}\mathbf{O}_{i,a}\mathbf{L}_{a,j}^{\top} = \sum_{a=1}^{i} 0 \times \mathbf{L}_{a,j}^{\top} + \sum_{a=i+1}^{j}\mathbf{O}_{i,a}\mathbf{L}_{a,j}^{\top} + \sum_{a=j+1}^{n}\mathbf{O}_{i,a} \times 0 = \sum_{a=i+1}^{j}\mathbf{O}_{i,a}\mathbf{L}_{a,j}^{\top} = \mathbf{P}_{i,j}
\tag{39}
$$

Thus, we have $\mathbf{P} = \left((\Delta\mathbf{X}\mathbf{X}^{\top}\mathbf{L}) \odot \mathbf{M}_{\mathbf{U}}\right)\mathbf{L}^{\top}$.

$\qquad\square$

## B. Additional Experiments

### B.1. Weight-only Quantization with Rotation

We tested our method for weight-only quantization on LLaMA2 and LLaMA3 models. The weight is quantized to 2/3/4 bits in per-channel asymmetric format. The compared baseline includes AWQ (Lin et al., 2023), OmniQuant (Shao et al.,

---

**Algorithm 2** GPTAQ quantization for entire transformer model

---

**Input:** Full-Precision Model, which contains $b$ blocks, two model input $\mathbf{X}$ and $\tilde{\mathbf{X}}$, AQ=True/False.
**for** $i = 1, 2, 3, \ldots, b$-th block **do**
    Move $block[i]$ to GPU memory                                                 *# Only 1 block is loaded into GPU*
    Disable activation quantization, if any
    $\tilde{\mathbf{X}} \leftarrow block[i](\tilde{\mathbf{X}})$, store FP input for each layer                       *# update FP block input data*
    **if** AQ is True **then**
        Enable activation quantization, if any                 *# Enable activation quantization during calibration*
    **end if**
    **for** $j = i, 2, \ldots, l$-th layer **do**
        $\_ = block[i](\mathbf{X})$, store input for $j$-th layer       *# Compute $\mathbf{H}$ and $\Delta\mathbf{X}\mathbf{X}^\top$ and delete FP input for this layer*
        Perform GPTAQ algorithm for $j$-th layer
        Quantize $j$-th layer's weight
    **end for**
    $\mathbf{X} \leftarrow block[i](\mathbf{X})$                                             *# update quantized block input data*
    Move $block[i]$ to CPU memory
**end for**

---

2023), QuaRot (Ashkboos et al., 2024), and QuaRot+GPTQ (Frantar et al., 2022). As demonstrated in the following table, QuaRot+GPTAQ is able to enhance the performance of weight-only quantization, too. This result demonstrates that even weight-only quantization will introduce residual output error in the quantized model. In W2A16 cases, this will significantly impact the GPTQ performance and our method can reduce the perplexity by $\sim$50%.

*Table 7.* Weight quantization results of language transformer. We report Wikitext2 perplexity.

| Precision | Method | FT-Free | L3-8B | L3-70B | L2-7B | L2-13B | L2-70B |
|---|---|---|---|---|---|---|---|
| FP16 | Pretrained | ✓ | 6.14 | 2.85 | 5.47 | 4.88 | 3.32 |
| | AWQ | ✓ | 6.57 | 3.59 | 5.82 | 5.07 | 3.49 |
| | OmniQuant | ✗ | - | - | 5.74 | 5.02 | 3.47 |
| W4A16 | QuaRot | ✓ | 7.72 | 16.7 | 6.76 | 5.48 | 3.66 |
| | QuaRot+GPTQ | ✓ | 6.54 | 3.55 | 5.60 | 5.00 | 3.41 |
| | QuaRot+GPTAQ | ✓ | **6.46** | **3.35** | **5.54** | **4.96** | **3.40** |
| | AWQ | ✓ | 11.8 | 12.3 | 14.2 | 6.32 | 4.22 |
| | OmniQuant | ✗ | - | - | 6.58 | 5.58 | 3.92 |
| W3A16 | QuaRot | ✓ | 39.7 | 138 | 146 | 48.9 | 5.25 |
| | QuaRot+GPTQ | ✓ | 7.59 | 5.37 | 6.08 | 5.37 | 3.72 |
| | QuaRot+GPTAQ | ✓ | **7.25** | **4.70** | **5.86** | **5.20** | **3.66** |
| | AWQ | ✓ | 4.1e5 | 8.6e4 | 2.9e6 | 6.2e3 | 3.9e3 |
| | OmniQuant | ✗ | - | - | 37.4 | 17.2 | 7.81 |
| W2A16 | QuaRot | ✓ | 4.0e4 | 4.7e4 | 9.7e3 | 5.2e3 | 1.2e3 |
| | QuaRot+GPTQ | ✓ | 23.2 | 18.5 | 20.7 | 10.9 | 5.60 |
| | QuaRot+GPTAQ | ✓ | **13.4** | **10.7** | **9.02** | **7.72** | **5.18** |

## C. Memory Analysis

We additionally analyze the GPU memory required to perform our algorithm (Algorithm 1). We focus our analysis on two key components in our method: $\tilde{\mathbf{X}}$ and $\mathbf{P}$. We first make the observation that $\tilde{\mathbf{X}}$ is only involved in the computation of $\Delta\mathbf{X}\mathbf{X}^\top$. Thus, the memory required for storing $\tilde{\mathbf{X}}$ is temporary and can be safely released as soon as $\Delta\mathbf{X}\mathbf{X}^\top$ is computed. $\mathbf{P}$, on the other hand, needs to be kept in the GPU memory during the iterative quantization updates.

Though $\tilde{\mathbf{X}}$ only requires temporary GPU memory, due to the inherent large dimensionality of $k$, materializing $\tilde{\mathbf{X}}$ for the entire model can potentially cause the GPU to suffer from high peak memory usage. We thus utilize a GPU-friendly strategy to manipulate the temporary storage of $\tilde{\mathbf{X}}$. As shown in Algorithm 2, for each block that awaits calibration, only the $\tilde{\mathbf{X}}$ that associates with the block will be materialized in the GPU memory. Assuming the block consists of $l$ layers, the temporary memory required by $\tilde{\mathbf{X}}$ is bounded by the complexity of $\mathcal{O}(nkl)$. As an example, on LLaMA2-7B, our profiling results show that $\tilde{\mathbf{X}}$ only temporarily induces $\sim$ 12GB memory space. Note that since the complexity solely depends on the block size, our storage strategy for $\tilde{\mathbf{X}}$ is scalable with the growing model size. In practice, we can further offload these data to

CPU memory and only load them to GPU when needed, which we empirically find brings negligible overhead to the overall latency but further improves the GPU memory efficiency.

As for $\mathbf{P}$, thanks to the relatively smaller dimensionality of $m, n$, the memory storage requirement is very small. In Table 8, we detail the dimensionality of different matrices that are kept in GPU memory during the iterative updates (left part) and the one for each local update round within the column blocks. For concrete reference, Table 9 provides the runtime GPU memory requirements for calibrating individual layers within a block of LLaMA2-7B.

We would like to strengthen and clarify the fact that throughout the entire quantization process, only one single copy of the model block exists inside the GPU memory. The $\tilde{\mathbf{X}}$ that contains the FP information of the block is first calculated by forwarding the input through the unquantized block and stored in the CPU memory. Then the same model block inside the GPU memory is quantized and calibrated by our proposed algorithm. As shown in Algorithm 2, the extra memory storage required by our method is solely brought by $\tilde{\mathbf{X}}$ and $\mathbf{P}$ which we have analyzed above.

*Table 8.* Dimensions of matrices used in GPTQ/GPTAQ.

| Method | $\mathbf{W}$ | $\mathbf{L}$ | $\mathbf{Q}$ | $\mathbf{E}$ | $\mathbf{P}$ | $\mathbf{W}_{:,Q}$ | $\mathbf{L}_{Q,Q}$ | $\mathbf{Q}_{:,Q}$ | $\mathbf{E}_{:,Q}$ | $\mathbf{P}_{Q,Q}$ |
|---|---|---|---|---|---|---|---|---|---|---|
| GPTQ | $m \times n$ | $n \times n$ | $m \times n$ | $m \times n$ | $0 \times 0$ | $m \times B$ | $B \times B$ | $m \times B$ | $m \times B$ | $0 \times 0$ |
| GPTAQ | $m \times n$ | $n \times n$ | $m \times n$ | $m \times n$ | $n \times n$ | $m \times B$ | $B \times B$ | $m \times B$ | $m \times B$ | $B \times B$ |

*Table 9.* Memory needed to perform calibration on LLaMA2-7B. $B = 128$ follows the standard setup.

| Layer | q_proj | k_proj | v_proj | o_proj | up_proj | gate_proj | down_proj |
|---|---|---|---|---|---|---|---|
| $m \times n$ | $4096 \times 4096$ | $4096 \times 4096$ | $4096 \times 4096$ | $4096 \times 4096$ | $11008 \times 4096$ | $11008 \times 4096$ | $4096 \times 11008$ |
| GPTQ | 0.13GB | 0.13GB | 0.13GB | 0.13GB | 0.29GB | 0.29GB | 0.48GB |
| GPTAQ | 0.16GB | 0.16GB | 0.16GB | 0.16GB | 0.32GB | 0.32GB | 0.70GB |

