# OpenReview forum: "GPTAQ: Efficient Finetuning-Free Quantization for Asymmetric Calibration"
_ICML.cc/2025/Conference — ICML 2025 poster_

### Official Review · Reviewer_aj2m · 2025-03-10

**Overall Recommendation:** 4

**Summary:**

Following the widely used quantization framework GPTQ, this work identifies the problem in GPTQ named symmetric calibration that emerges from the per-layer optimization scheme. To tackle these challenges, this work proposes a unique calibration pipeline based on asymmetric calibration, which fully considers the quantization error and deviation in the output when updating the weights. Concretely, channel parallelization, neuron decomposition, and Cholesky reformulation for matrix fusion are utilized to parallelize the solution. The proposed GPTQv2 is extensively verified across various LLMs and ViTs on multiple tasks, demonstrating remarkable efficiency and effectiveness. It can be a plugin for QuaRot/SpinQuant and improve the performance with minimal overhead.

**Claims And Evidence:**

Most claims made in the submission are supported by clear and convincing evidence.

**Essential References Not Discussed:**

Existing work related to quantization and Optimal Brain Surgeon (OBS) are cited.

**Experimental Designs Or Analyses:**

I have checked the soundness/validity of experimental designs and analyses. Some issues include:
- Bitwidth settings are limited to W4A4/W2A4. Considering the need for near-lossless quantization of LLMs/ViTs in some application scenarios, W6A6/W8A8 results would be a plus.
- Missing baselines for ViT quantization. Rotation-based methods such as QuIP can also be applied in ViT quantization, and FrameQuant[1] is omitted.
- Figure 4 (a) seems to be missing bars for latency comparison. The latency overhead of GPTQv2 compared to GPTQ is non-negligible, especially on higher dimensions.
- For SpinQuant, I think it should be categorized as FT-free since it only involves optimizing the rotation but not updating the weights. In addition, the author of SpinQuant mentioned that optimizing rotation is highly efficient, 0.5h for L3-8B in their paper. It could be the difference in hardware system, please double check it.
- It would be interesting to see if the proposed GPTQv2 could be applied to Diffusion Transformers such as Q-DiT [2]. (No need to perform additional experiments, discussion on it would be enough)

[1] FrameQuant: Flexible Low-Bit Quantization for Transformers, ICML 2024

[2] Q-DiT: Accurate Post-Training Quantization for Diffusion Transformers, arxiv 2023/CVPR 2025

**Methods And Evaluation Criteria:**

The proposed methods and evaluation criteria make sense for the problem. Algorithm1 gives a clear introduction to how GPTQv2 differs from GPTQ. The proposed techniques are well-motivated.

**Other Comments Or Suggestions:**

N/A

**Other Strengths And Weaknesses:**

N/A

**Questions For Authors:**

Please see the points above.

**Relation To Broader Scientific Literature:**

This work is built on GPTQ, but the core contribution is novel.

**Theoretical Claims:**

check the correctness of proofs for theoretical claims in the main text.

---

> ### Author Rebuttal · Authors · 2025-03-29
>
> Thanks for your comments and positive feedback. Please check our response below.
>
> >1. Bitwidth settings are limited to W4A4/W2A4. Considering the need for near-lossless quantization of LLMs/ViTs in some application scenarios, W6A6/W8A8 results would be a plus.
>
> Thank you for this suggestion. We have conducted additional experiments with W6A6 bitwidth on LLaMA2-7B and LLaMA3-8B to evaluate GPTQv2 in near-lossless quantization scenarios. The results are presented below (reporting WikiText-2 perplexity):
>
> | Model             | LLaMA2-7B | LLaMA3-8B |
> |-------------------|-----------|-----------|
> | Pretrained (FP16) | 5.47      | 6.14      |
> | OmniQuant         | 5.87      | 7.24      |
> | QLLM              | 5.91      | -         |
> | DuQuant           | 5.53      | 6.27      |
> | QuaRot+GPTQ       | 5.50      | 6.24      |
> | QuaRot+GPTQv2     | 5.49      | 6.21      |
>
>
> As expected, the improvements in the W6A6 setting are more modest compared to those in lower bitwidth scenarios, since higher precision quantization already preserves most of the model's capabilities, leaving less room for enhancement. Nevertheless, GPTQv2 still consistently outperforms GPTQ across both models.
>
> >2. Missing baselines for ViT quantization. Rotation-based methods, such as QuIP, can also be applied in ViT quantization, and FrameQuant[1] is omitted.
>
> Thanks for letting us know about the FrameQuant paper. As far as we can tell, QuIP and FrameQuant are performing weight-only quantization on ViTs, which would be less effective than LLM decoding, as the ViT inference is compute-bounded instead of I/O-bounded.
>
> Nevertheless, we test DeiT-S with 2-bit per-channel quantization with or without QuIP incoherence processing. The results are shown below, with * indicating our implementation.
>
> | Method             | ImageNet accuracy |
> |--------------------|-------------------|
> | FrameQuant (r=1.0) | 66.35             |
> | QuIP               | 65.70             |
> | GPTQ*              | 57.11             |
> | GPTQv2*            | 60.58             |
> | QuIP + GPTQ*       | 65.45             |
> | QuIP + GPTQv2*     | 68.02             |
>
>
> >3. Figure 4 (a) seems to be missing bars for latency comparison. The latency overhead of GPTQv2 compared to GPTQ is non-negligible, especially on higher dimensions.
>
> Sorry for the missing histograms. This appears to be a browser-specific rendering problem. If you download our paper, the histograms should be visible when viewing the PDF in Chrome or in standard PDF readers. We will ensure the figure is displayed correctly in all formats in the revised version of the paper.
>
> For the GPTQv2 vs GPTQ latency comparison, we can give a theoretical upper bound here. According to Algorithm 1, the operations of v2 are at most 2x of v1. With sufficiently high dimensions, we can expect the latency may approach 2x. However, in practice, we observe 30%-50% more time in typical LLMs (7B to 405B).
>
> >4. For SpinQuant, I think it should be categorized as FT-free since it only involves optimizing the rotation but not updating the weights. In addition, the author of SpinQuant mentioned that optimizing rotation is highly efficient, 0.5h for L3-8B in their paper. It could be the difference in hardware system, please double check it.
>
> We maintain that SpinQuant should be categorized as finetuning-based rather than finetuning-free for the following reasons:
>
> + From a computational perspective, optimizing rotation matrices or weights involves the same core operations - both require backpropagation through the network, use of the Straight-Through Estimator (STE), and gradient-based optimization. The fact that the optimization target is a rotation matrix rather than weight values does not reduce the computational requirements.
>
> + Conceptually, SpinQuant modifies the effective weight representation through rotation optimization. Whether directly updating weights or optimizing transformations applied to weights, both approaches adjust the model's parameters that will be quantized.
>
> Regarding the finetuning time, it’s due to the SpinQuant authors using 8 A100 GPUs to finetune the rotation matrix. Therefore, in Section 5.3, we explained that “we additionally report the GPU Hours (on one A100) required to run the algorithm. Although in practice, SpinQuants runs on 8 A100 GPUs ”
>
> >5. It would be interesting to see if the proposed GPTQv2 could be applied to Diffusion Transformers such as Q-DiT [2].
>
> Thanks for the reference on transformer-based diffusion models. We noticed that Q-DiT pointed out an important observation that activation distribution undergoes continuous changes across timesteps. In this case, the activation asymmetry may accumulate not just through layers but through time steps as well. We expect GPTQv2 will have better performance if $\Delta \mathbf{X}$ can capture more information as we did in the experiments of quantization order (Appendix B.1). We will discuss this problem of diffusion models in our paper related work.

---

> > ### Comment · Reviewer_aj2m · 2025-04-04
> >
> > Thanks for the response from the authors. The additional results on W6A6 and ViTs are helpful, please consider including them in the revised draft. Most of my concerns are well-addressed, therefore, I would like to increase my score to 4.

---

> > > ### Author Response · Authors · 2025-04-04
> > >
> > > Thank you very much for raising the score. We will add the new results to our final version of the paper.

---

### Official Review · Reviewer_fFrA · 2025-03-11

**Overall Recommendation:** 4

**Summary:**

Authors propose a novel fine-tuning free quantization framework GPTQv2 for LLMs. Authors first  analyze the problem of previous "symmetric calibration"using optimal brain compression to derive a close-formed solution, and propose a novel "asymmetric calibration" to take  quantization error as well as the accumulated asymmetry error into consideration. Secondly, authors utilize various techniques to parallelize the solution calculation, including channel parallelization, neuron decomposition, and Cholesky reformulation for matrix fusion. Extensive results on various LLMs and datasets reveal the effectiveness of the proposed methods.

**Claims And Evidence:**

All claims are well-explained.

**Essential References Not Discussed:**

All necessary references are discussed.

**Experimental Designs Or Analyses:**

I've checked all experimental settings, comparison and results in this paper. See "Other Strengths And Weaknesses" part of this review for my major & minor concerns about the experimental part.

**Methods And Evaluation Criteria:**

I've checked all theoretical and qualitative analysis and claims in this paper. See "Other Strengths And Weaknesses" part of this review for my major & minor concerns about the methodology and equation derivation.

**Other Comments Or Suggestions:**

See above.

**Other Strengths And Weaknesses:**

## Major weakness
1. What does the lambda in eq.10 represent for? Is it a hyper-parameter? If so, then why there is a gradient on it?
2. In table. 2, why the performance improvement is more significant on fine-tuning quantization methods than fine-tuning free ones, when compared with GPTQ? It would be better to deeply discuss about the phenomenon.
3. In table. 4, why the ppl result of GPTQv2' is worse than GPTQ, while the zero-shot avg result is better than GPTQ, which is counter-intuitive.
4. I curious about the performance improvement when activation quantization is added before weight quantizations. It seems like the improvement compared to GPTQ is more significant under this condition. In my perspective, the quantized  $\\tilde{X}$ will introduce noise into $\\Delta X$, therefore, it should be worse.

## Minor weakness
1. Seems like the histograms in Figure 4(a) are missing.

**Questions For Authors:**

See above.

**Relation To Broader Scientific Literature:**

All contributions are technical and all datasets used for experiments are open-sourced. Thus no key contributions of this paper related to the broader scientific literature.

**Theoretical Claims:**

I've checked all theoretical and qualitative analysis and claims in this paper. See "Other Strengths And Weaknesses" part of this review for my major & minor concerns about the methodology and equation derivation.

---

> ### Author Rebuttal · Authors · 2025-03-29
>
> Thank you very much for your comments and thorough review. Please check our response to your questions.
>
> >1. What does the lambda in eq.10 represent for? Is it a hyper-parameter? If so, then why there is a gradient on it?
>
> $\lambda$ is the Langrange multiplier, which is not a hyperparameter. By taking derivatives of the Lagrangian with respect to both $\Delta\mathbf{w}$ and $\lambda$ and setting them to zero, we simultaneously: (1) ensure the quantization constraint is satisfied exactly ($\partial L/\partial \lambda=0$ enforces $\Delta\mathbf{we}_q^\top+\mathbf{w}_q-\hat{\mathbf{w}}_q=0$), and (2) find the local minimum of the objective function. This approach follows standard constrained optimization techniques used in the original OBS and OBQ frameworks.
>
> >2. In table. 2, why the performance improvement is more significant on fine-tuning quantization methods than fine-tuning free ones, when compared with GPTQ? It would be better to deeply discuss about the phenomenon.
>
> We think the reason why finetuning quantization performs worse is the need to handle the massive outliers in activations. This issue was addressed only with QuaRot (FT-free) and SpinQuant (FT-based) until recently.
>
> >3. In table. 4, why the ppl result of GPTQv2' is worse than GPTQ, while the zero-shot avg result is better than GPTQ, which is counter-intuitive.
>
> This is an excellent observation. Perplexity and zero-shot accuracy measure different aspects of LLM capabilities. Perplexity primarily evaluates next-token prediction on the pre-training distribution (a form of memorization), while zero-shot accuracy tests the model's ability to generalize knowledge to new tasks.
> When applying only the second term of our method, we're optimizing for the residual output error from previous layers, which better preserves the model's generalization capabilities at the expense of exact next-token prediction. The full GPTQv2 balances both aspects by combining both terms. This suggests that different quantization objectives might be optimal depending on the downstream task priorities.
>
>
> >4. I curious about the performance improvement when activation quantization is added before weight quantizations. It seems like the improvement compared to GPTQ is more significant under this condition. In my perspective, the quantized $\tilde{\mathbf{X}}$ will introduce noise into $\Delta \mathbf{X}$, therefore, it should be worse.
>
> Thanks for the question. We kindly refer you to our Algorithm 2 in Appendix C. If the activation quantization is enabled during calibration, we will disable it when computing and caching $\tilde{\mathbf{X}}$ to ensure we always use the FP model activation. This is why the improvement will be more significant.
>
>
> >5. Seems like the histograms in Figure 4(a) are missing.
>
> Sorry for the missing histograms. This appears to be a browser-specific rendering problem. If you download our paper, the histograms should be visible when viewing the PDF in Chrome or in standard PDF readers. We will ensure the figure is displayed correctly in all formats in the revised version of the paper.

---

### Official Review · Reviewer_vtou · 2025-03-12

**Overall Recommendation:** 4

**Summary:**

The authors introduce a new quantization method, GPTQv2. The key innovation here is the development of an asymmetric calibration approach, differing fundamentally from GPTQ, by explicitly aligning the quantized layer's outputs to the original, full-precision activations. They derive a closed-form solution using Optimal Brain Compression principles. Experiment evaluations show substantial improvements in model performance across vision and language tasks.

## Update after rebuttal
I maintain my original score. I am generally satisfied with the authors’ response.

**Claims And Evidence:**

The paper claims to achieve superior performance as compared to GPTQ and supports its claims via various experiments.

**Essential References Not Discussed:**

NA

**Experimental Designs Or Analyses:**

Experiments and ablation studies seem methodologically sound and thorough. However, actual hardware numbers would strengthen the work further.

**Methods And Evaluation Criteria:**

Chosen evaluation metrics (e.g., perplexity, accuracy on PiQA, HellaSwag, etc.) are appropriate and standard for the field.

**Other Comments Or Suggestions:**

NA

**Other Strengths And Weaknesses:**

**Strengths**
- Clear and thorough theoretical foundation with detailed mathematical derivations provided.
- Efficient computational strategies that significantly reduce quantization overhead, making practical implementation feasible.
- Extensive experiments demonstrating clear and consistent performance improvements across different transformer architectures and tasks.

**Weaknesses**:

- Lack of detailed hardware-level deployment and overhead analyses slightly limits practical applicability insights.

**Questions For Authors:**

NA

**Relation To Broader Scientific Literature:**

GPTQv2 is effectively contextualized against existing methods, clearly highlighting its innovative elements over the original GPTQ algorithm and other finetuning-free quantization methods.

**Theoretical Claims:**

Theoretical claims sound reasonable, addressing the shortcomings of the original GPTQ work.

---

> ### Author Rebuttal · Authors · 2025-03-29
>
> Thank you for your positive assessment of our theoretical foundation and experimental results. Regarding your concern about hardware-level deployment and overhead analyses, we would like to clarify that GPTQv2 maintains full compatibility with GPTQ's quantization format since we did not modify the `quant()` function in Algorithm 1. This means that our method can leverage all existing hardware-optimized kernels and infrastructure developed for GPTQ without additional overhead during inference.
> Take an example of the format of [`AutoGPTQ`](https://github.com/AutoGPTQ/AutoGPTQ) library, for every quantized layer, the variables are defined as
>
> ```python
> class QuantLinear(nn.Module):
>     def __init__(self, bits, group_size, in_features, out_features):
>         self.bits = bits
>         self.group_size = group_size
>         m, n = in_features, out_features
>         self.qweight = torch.zeros((n, m//32 * self.bits), dtype=torch.int32)
>         self.qzeros = torch.zeros((n//group_size, m//32 * self.bits), dtype=torch.int32)
>         self.scales = torch.zeros((n//group_size, m), dtype=torch.float16)
>         self.g_idx = torch.zeros(n, dtype=torch.int32)
> ```
>
> The same quantization format in GPTQv2 can immediately benefit from specialized kernels like [Marlin](https://github.com/IST-DASLab/marlin) and [ExLLaMA](https://github.com/turboderp-org/exllamav2) without requiring new hardware optimizations. Currently, we are integrating GPTQv2 into popular quantization libraries, and we will expand our hardware-specific analyses in the next version of the paper.

---

### Official Review · Reviewer_3TqR · 2025-03-13

**Overall Recommendation:** 4

**Summary:**

This paper proposed a modification to the widely-used GPTQ method. The main idea is that instead of minimizing the differences between quant(W)*A and W*A, authors proposed to minimize the differences between quant(W)*A with W*A_fp, i.e. its counterpart in the unquantized model. As in typical PTQ works, "sequential" quantization, i.e. assuming layer 0 to l-1 are quantized while quantizing layer l, is generally considered more effective, because later layers may have a chance to absorb some quantization errors accumulated from quantizing previous layers. Directly matching layer output to counterparts in unquantized model would be closer to the idea of distillation, which usually requires more iterations and data to achieve better results. Interestingly this work showed that for GPTQ, "distillation style" would be a better option than "sequential PTQ style."

**Claims And Evidence:**

Yes

**Essential References Not Discussed:**

Citations/references are sufficient.

**Experimental Designs Or Analyses:**

Yes, the choice of models include vision transformers and LLMs, model size ranged from <1B, 7-70B, and 405B is proper. The selection of metrics include ImageNet accuracy, wiki2 perplexity, and a range of 0-shot tasks is valid.

**Methods And Evaluation Criteria:**

Yes

**Other Comments Or Suggestions:**

please see Weakness above

**Other Strengths And Weaknesses:**

Strength:
1. Well written manuscript.
2. Good amount of experimental data, including those additional results in appendix. Achieved meaningful improvement in accuracy/perplexity compared to original GPTQ.
3. Considered implementation efficiency and provide improved formulas so that overall process time would be comparable to original GPTQ.

Weakness:
overall, very nice work. Just a few minor suggestions.
1. The terms "asymmetric calibration" and "symmetric calibration" doesn't seem to be very intuitive, and maybe a bit confusing with the symmetric/asymmetric quantization. In fact, this set of terms is not used a lot in the manuscript. Maybe author can consider adding a few sentences around the definition of these terms to enhance the connection between the main proposed concept, so that the readers could grasp the main idea easier.

2. Since this work is meant to be an improvement of the original GPTQ, it would be beneficial to start the discussion with a comparison to vanilla GPTQ, i.e. weight only, per-group quantization. Maybe author could consider moving Appendix B.3/Table 7 into main manuscript, with a few more examples from different Llama models.

3. Even though (possibly) the majority of researchers still calls it GPTQ, the original author officially published their work at ICLR 2023 in the name of "OPTQ". I would not suggest the author of this work to change all the names/acronyms in the manuscript, but in respect of the choice of the original "GPTQ" authors, maybe include both names during first mention/citation and state that for only GPTQ will be used afterward for simplicity reason.

**Questions For Authors:**

None

**Relation To Broader Scientific Literature:**

This work is an improvement of a widely-used method, GPTQ, which is a weight-only quantization method that can address memory-bound issue of LLMs.

**Theoretical Claims:**

Yes, Appendix A.1, seems fine.

---

> ### Author Rebuttal · Authors · 2025-03-29
>
> Thank you for your positive feedback on our manuscript. We appreciate your interpretation of GPTQ as a "sequential PTQ style" method versus the "distillation style" of our GPTQv2. Please see our responses to your specific concerns below:
>
> >1. The terms "asymmetric calibration" and "symmetric calibration" doesn't seem to be very intuitive, and maybe a bit confusing with the symmetric/asymmetric quantization. In fact, this set of terms is not used a lot in the manuscript. Maybe author can consider adding a few sentences around the definition of these terms to enhance the connection between the main proposed concept, so that the readers could grasp the main idea easier.
>
> We agree that our terminology could be clearer. The terms "symmetric" and "asymmetric" specifically refer to the calibration objective: symmetric calibration uses the same input activation $\mathbf{X}$ for both the quantized and full-precision weights, while asymmetric calibration accounts for the input activation discrepancy between quantized weights and full-precision weights. We will add clearer definitions of these terms in the introduction and highlight their conceptual differences.
>
> >2. Since this work is meant to be an improvement of the original GPTQ, it would be beneficial to start the discussion with a comparison to vanilla GPTQ, i.e. weight only, per-group quantization. Maybe author could consider moving Appendix B.3/Table 7 into main manuscript, with a few more examples from different Llama models.
>
> Thank you for your suggestion. We agree that demonstrating improvements over vanilla GPTQ (weight-only, per-group quantization) is important since many existing libraries target this setup. In the revised version, we will move Table 7 to the main text and expand it with comprehensive comparisons across multiple LLaMA models. Here are additional results for LLaMA3-8B's perplexity:
>
> | Bitwidth | W4A16-G128 | W3A16-G128 | W2A16-G128 |
> |----------|------------|------------|------------|
> | GPTQ     | 6.71       | 7.91       | 25.24      |
> | GPTQv2   | 6.41       | 7.72       | 14.17      |
>
>
>
> >3. Even though (possibly) the majority of researchers still calls it GPTQ, the original author officially published their work at ICLR 2023 in the name of "OPTQ". I would not suggest the author of this work to change all the names/acronyms in the manuscript, but in respect of the choice of the original "GPTQ" authors, maybe include both names during first mention/citation and state that for only GPTQ will be used afterward for simplicity reason.
>
> This is an essential problem that we were not aware of. We'll add a footnote in the introduction to clarify the naming issue. Thanks again for your suggestion.

---

> > ### Comment · Reviewer_3TqR · 2025-04-03
> >
> > Thank for your clarifications. I would like to keep my assessment unchanged.

---

> > > ### Author Response · Authors · 2025-04-04
> > >
> > > Thank you very much for your positive feedback. We will clarify them in the final version.

---

### Decision · Program_Chairs · 2025-05-01

**Decision:**

Accept (poster)

**Comment:**

This paper enhances the widely used GPTQ approach, with its primary novelty being the development of an asymmetric calibration method to align the quantized layer's outputs with full-precision activations. The work achieves significant improvements across benchmarks and provides extensive evaluation. AC agrees with all reviewers in recommending acceptance of this paper.